# Ton motor conformational switch and peptidoglycan role in bacterial nutrient uptake

Maximilian Zinke[1], Maylis Lejeune [1], Ariel Mechaly[2], Benjamin Bardiaux [1], Ivo Gomperts Boneca [3], Philippe Delepelaire[4,5] & Nadia Izadi-Pruneyre [1] ✉

Active nutrient uptake is fundamental for survival and pathogenicity of Gram-negative bacteria, which operate a multi-protein Ton system to transport essential nutrients like metals and vitamins. This system harnesses the proton motive force at the inner membrane to energize the import through the outer membrane, but the mechanism of energy transfer remains enigmatic. Here, we study the periplasmic domain of ExbD, a crucial component of the proton channel of the Ton system. We show that this domain is a dynamic dimer switching between two conformations representing the proton channel's open and closed states. By in vivo phenotypic assays we demonstrate that this conformational switch is essential for the nutrient uptake by bacteria. The open state of ExbD triggers a disorder to order transition of TonB, enabling TonB to supply energy to the nutrient transporter. We also reveal the anchoring role of the peptidoglycan layer in this mechanism. Herein, we propose a mechanistic model for the Ton system, emphasizing ExbD duality and the pivotal catalytic role of peptidoglycan. Sequence analysis suggests that this mechanism is conserved in other systems energizing gliding motility and membrane integrity. Our study fills important gaps in understanding bacterial motor mechanism and proposes novel antibacterial strategies.

Gram-negative bacteria present a unique challenge for the development of novel drugs due to their dual-membrane structure, which effectively protects them by preventing many antibiotics from accessing their targets within the cell[1]. This dual-membrane architecture requires specialized transport systems for essential nutrients – like iron, nickel, vitamin B12 and certain carbohydrates[2]. These gram-negative specific systems guarantee efficient transport over both the inner and the outer membrane as well as the periplasm, potentially creating vulnerabilities for therapeutic intervention. One of the key systems involved in this transport is the Ton system, which forms a multi-protein complex embedded in the inner membrane (Fig. 1a). This system utilizes the proton motive force (PMF) to physically open a variety of outer membrane transporters – the so-called TonB-dependent transporters (TBDTs) – and, hereby, realizes active transport by means of the inner membrane proton channel-forming complex ExbB-ExbD and the periplasm crossing protein TonB. While the TBDTs are nutrient-specific, TonB and ExbB-ExbD can be employed to multiple TBDTs (multi-target)[3]. In addition to this multi-target Ton system, some Ton systems are dedicated to a specific transporter (single-target). This is the case of the Heme Acquisition System (Has) allowing bacteria to acquire heme as an iron source and involving orthologs of ExbB and ExbD with a TonB paralog called HasB[4–6].

At the inner membrane, the Ton system consists of the proton-channel forming complex ExbB-ExbD and the protein TonB, which are

[1]Institut Pasteur, Université Paris Cité, CNRS UMR3528, Bacterial Transmembrane Systems Unit, F-75015 Paris, France. [2]Institut Pasteur, Université Paris Cité, CNRS UMR3528, Crystallography Platform, F-75015 Paris, France. [3]Institut Pasteur, Université Paris Cité, CNRS UMR6047, INSERM U1306, Unité de Biologie et génétique de la paroi bactérienne, F-75015 Paris, France. [4]Laboratoire de Biologie Physico-Chimique des Protéines Membranaires, Université Paris Cité, UMR7099 CNRS, F-75005 Paris, France. [5]Institut de Biologie Physico-Chimique, F-75005 Paris, France. ✉e-mail: nadia.izadi@pasteur.fr

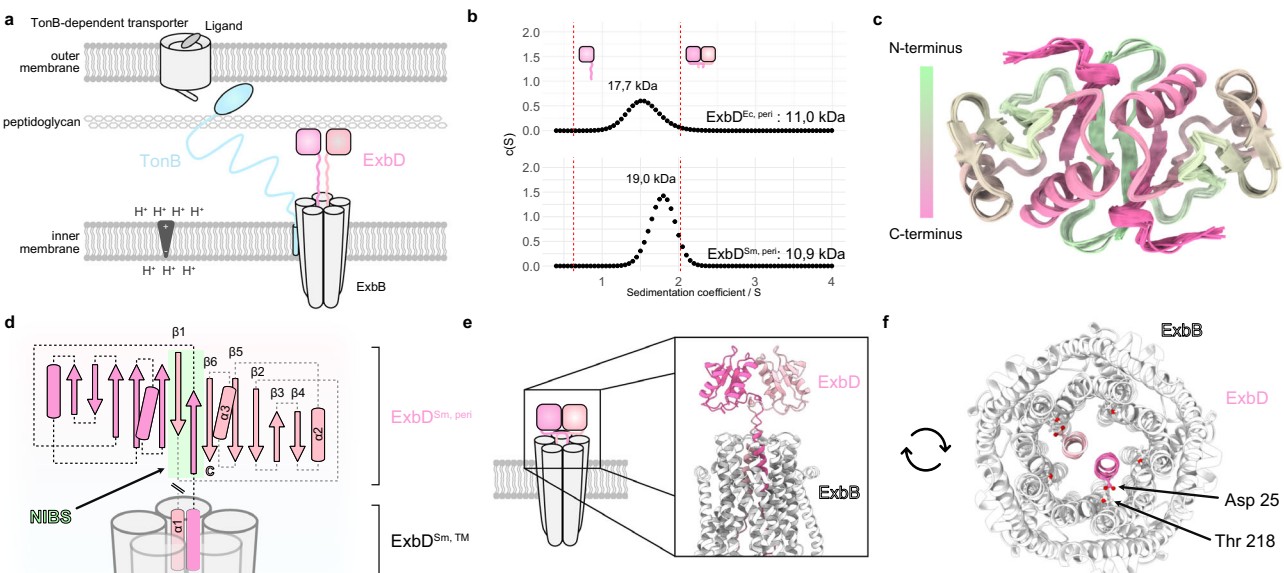

**Fig. 1 | The periplasmic domain of ExbD forms a homodimer. a** The Ton system consists of an inner membrane proton channel – formed by ExbB (grey) and ExbD (pink) – and the periplasm spanning protein TonB (blue). The latter links the system to a TonB-dependent transporter in the outer membrane. The globular C-terminal domain of ExbD (pink boxes) is connected to its helical N-terminus, which is inserted into the ExbB channel, by a disordered region (pink, wiggly lines). **b** Sedimentation coefficient distribution obtained from analytical ultracentrifugation of ExbD$^{Sm, peri}$ and ExbD$^{Ec, peri}$ indicates that both proteins exist in a dimeric state at physiological pH. The red dashed lines represent estimated sedimentation coefficients for monomeric ExbD and dimeric ExbD as depicted by the cartoons (pink). The sedimentation coefficient-derived molecular weights of both species are indicated above. **c** The top view of the NMR structure ensemble of dimeric ExbD$^{Sm, peri}$ reveals that the N-terminal residues 44–49 (green) form an inter-molecular β-sheet. **d** Homodimeric ExbD$^{Sm, peri}$ consists of two monomers (pink and light pink), each with a β-sheet (arrows) and two α-helices (barrels). Part of the dimeric interface is formed by a swapped, intermolecular, and anti-parallel β-sheet called the N-terminal Intermolecular Beta-Strand (NIBS, green). The NIBS, composed of residues 44-49, connects the intramolecular β-sheets, creating a continuous β-sheet across the dimer interface. In the full-length protein, ExbD$^{Sm, peri}$ is N-terminally connected to two α-helices (α1) that are embedded in the ExbB channel (grey). **e** Side view and **f** Top view of the AlphaFold2 model of the full ExbB-ExbD complex, generated using the NMR structure (PDB ID 8PEK structure of the dimeric, periplasmic domain of ExbD, this work) as a template. In this model, the dimeric organization of the periplasmic domain of ExbD (pink and light pink) imposes the alignment of its helices in similar positions and orientations, and locking both Asp 25 sidechains in hydrogen bonds with Thr 218 from ExbB. In the top view, the periplasmic domain of ExbD is not visible due to its location above the clipping plane. Source data are provided as a Source Data file.

together capable of harnessing the PMF. The structure of the whole complex is unknown. Recent cryo-EM data showed two α-helices situated in a periplasm-facing central pore of the pentameric ExbB channel. These densities were attributed to the most N-terminal part of an ExbD dimer (residues 12-42)[7]. However, no information about the organization of the C-terminal, periplasmic part of ExbD could be gained as it remains invisible in all known cryo-EM structures to date, most likely due to its dynamics[5,8,9]. The only structural data is a monomeric, solution NMR structure of the periplasmic domain of ExbD at pH 3 that features an unfolded N-terminus (residues 44–63) followed by a globular fold (residues 64–132)[10]. It has been proposed that a PMF-dependent de- and reprotonation of the highly conserved ExbD residue Asp 25 in the helix that is embedded in the pore of the ExbB channel, causes a rotation of ExbD within ExbB[11] – analogous to the MotA-MotB system involved in flagellar motility[12]. At the outer membrane, ligand binding to a TBDT results in the propulsion of an N-terminal extension of the TBDT into the periplasm, where it is tightly bound to TonB via a conserved region called the TonB box[13]. TonB consists of an N-terminal α-helix anchoring it into the inner membrane and proposedly to ExbB[5], a periplasm-spanning intrinsically disordered linker, and a C-terminal, globular domain responsible for TBDT binding[14]. The intrinsically disordered linker contains a proline-rich region, approximately 70 residues in length, with an N-terminal stretch of Glu-Pro repeats and a C-terminal stretch of Lys-Pro repeats[15]. These stretches are organized in polyproline II helices and behave like two static rods, which however still retain some flexibility[16,17]. It is proposed that following TonB's binding to the TonB box, the hypothesized ExbD rotation would generate a pulling force on TonB which unfolds a plug in the TBDT, opening a channel and ensuring selective

ligand entry[18]. These dynamics might vary in the Has system due to a couple of distinguishing features: (1) The disordered linker of HasB does not contain an extensive stretch of Glu-Pro repeats, consequently lacking two large regions of opposite charges. (2) HasB is likely perpetually bound to the TBDT, even in the absence of the ligand. This characteristic can be attributed to the single-target nature of the HasB, eliminating the necessity to accommodate multiple TBDTs[19].

Due to the partial localization of the Ton system in the periplasm, a potential role of the periplasmic peptidoglycan layer in the mechanism of the system has been proposed[20]. In the related MotA-MotB system, MotB contains a C-terminal peptidoglycan-binding motif that anchors the MotAB stator to the cell wall[21], while in the analogous Tol-Pal system[22], this interaction is facilitated by Pal and TolR[23,24]. However, the nature of this interaction in the Ton system has remained elusive.

Despite being first identified in 1978[25], key steps of the Ton system's mechanism remain poorly understood primarily due to incomplete structural information, especially on the connection between ExbD and TonB. This is likely due to the disordered and dynamic nature of the periplasmic domains of both proteins, which renders them unsuitable for cryo-EM studies. Uncovering the role and the assembly of these two proteins is of great importance, as it could not only improve our understanding of nutrient transport in gram-negative bacteria but also provide new targets for antibiotics development. To ensure the broad relevance of our results, we investigated two representative systems: the multi-target Ton system from *Escherichia coli* and the single-target Has system from *Serratia marcescens*. These systems were chosen due to the plethora of existing functional data and a partial knowledge of their structures, allowing their

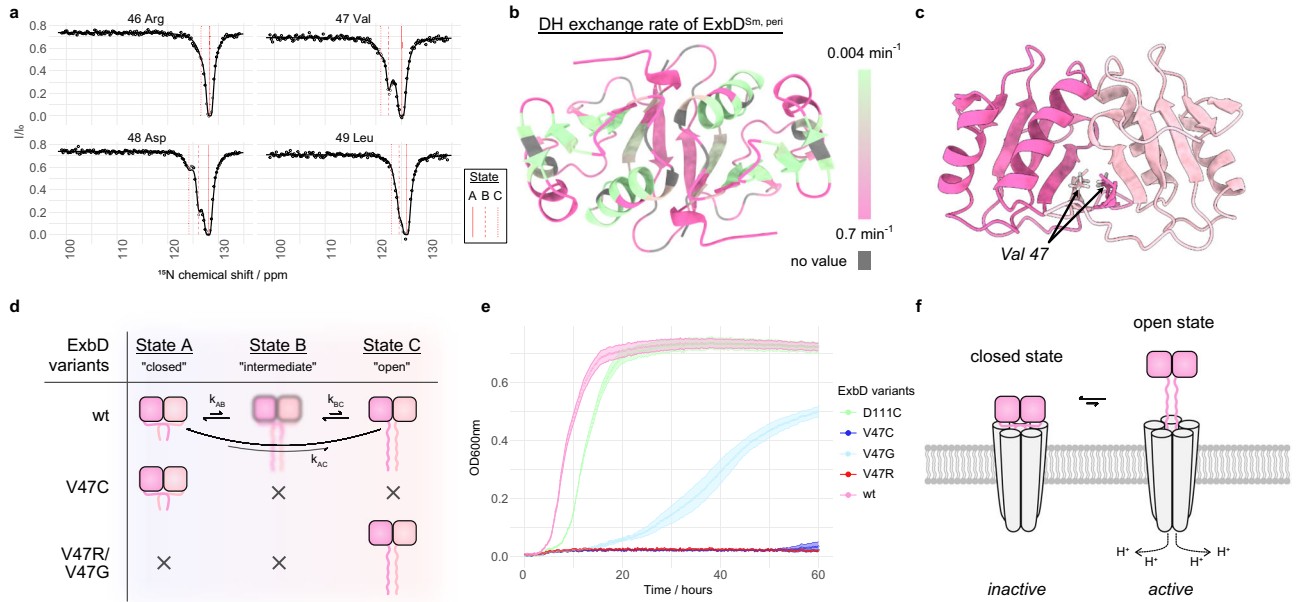

**Fig. 2 | The periplasmic domain of ExbD samples different conformational states. a** $^{15}$N-Chemical exchange saturation transfer (CEST) profiles reveal a 3-state exchange of the NIBS in $^{15}$N-labeled ExbD$^{Sm, peri}$. The experimental data points and the least-square fit are represented by dots and solid lines, respectively. The fitted chemical shift values for states A, B, and C are indicated by red lines. Intensity ratios of NMR signals from residues 46 to 49 are shown. Peak intensities are plotted as mean values with error bars showing the standard deviation (SD), which quantifies the relative measurement error arising from noise in the NMR spectra. **b** Deuterium hydrogen (DH) exchange rates measured by NMR spectroscopy mapped onto the structure of ExbD$^{Sm, peri}$ (fast exchange: pink, slow exchange: green, no value: grey) show that the NIBS undergoes faster amide proton exchange than adjacent β-strands, supporting the disordered nature of state C. **c** Side view of dimeric ExbD$^{Sm, peri}$ where the monomeric units are colored in pink and light pink. The residue Val 47 is located at the dimer interface in the closed state. **d** Schematic representation of the exchange dynamics of the ExbD variants. Wild type (wt) ExbD$^{Sm, peri}$ (pink) samples a major state A, and the minor states B and C. State A is characterized by a folded NIBS, while state C features a disordered NIBS. State B represents an intermediate state between these two extremes. The V47C mutant forms a disulfide bridge that locks ExbD$^{Sm, peri}$ in the closed state, while the V47R and V47G mutants mainly yield the open state. The pink rectangles depict the protomers of ExbD$^{Sm, peri}$. The intertwined pink lines symbolize the folded NIBS in the closed state, while the wiggly lines illustrate the disordered NIBS in the open state. **e** Bacterial growth curves of ExbD$^{Sm, peri}$ variants wt (pink), V47R (red), V47G (light blue), V47C (blue), and the control D111C (green). Under in vivo conditions that necessitate the activity of the Has system, these mutations substantially impact bacterial growth, emphasizing that not merely the individual states of ExbD, but the exchange between them, is vital for the system's proper functioning. Each data point in the growth curves represents the mean of the optical density for each variant, measured in triplicate ($n = 3$ biologically independent experiments). Error bars denote the SD from these replicates. **f** In a schematic representation of the ExbB-ExbD complex, the ExbD dimer exchanges between a closed, main state and a sparsely populated open state. The transition between the states is characterized by the unfolding and refolding of the NIBS residues. Herein, only the open state might be permeable to protons. Source data are provided as a Source Data file.

comprehensive analysis. In this study, we use NMR spectroscopy to make the dynamic visible: we present the dimeric structure of the periplasmic domain of ExbD in different states, including a sparsely populated one. We demonstrate that this minor state is conformationally selected upon binding to an intrinsically disordered region (IDR) of TonB, which undergoes a disorder-to-order transition. Moreover, mutagenesis and in vivo phenotypic assays confirm that this multi-state transition of ExbD is required for its function. Also, we show that the transient interaction of the ExbD dimer to the peptidoglycan layer in the periplasm is crucial for the action of the Ton system.

## Results

### The periplasmic domain of ExbD forms a homodimer

We recombinantly produced the periplasmic domains of ExbD from both organisms (from *E. coli* including residues 43-141: ExbD$^{Ec, peri}$, from *S. marcescens* including residues 43-140: ExbD$^{Sm, peri}$; all constructs and NMR samples are summarized in Supplementary Tables 1, 2). Analytical ultracentrifugation (AUC) revealed that the periplasmic domain of ExbD of both systems is organized as a dimer in solution (Fig. 1b). We proceeded with our structural study with ExbD$^{Sm, peri}$ as NMR fingerprint spectra indicated that ExbD$^{Ec, peri}$ suffers from line broadening due to conformational exchange dynamics, while the *S. marcescens* protein revealed high-quality spectral data (Supplementary Fig. 1, 2). The resonance assignment of ExbD$^{Sm, peri}$ yielded a sequence coverage of 100%, with at least one atom assigned in each residue. The solution

structure of ExbD$^{Sm, peri}$ was determined as detailed in the Methods and resulted in a swapped homodimer (Fig. 1c and Supplementary Table 3). Each protomer is composed of a four-stranded β-sheet on one side and two α-helices on the other (Fig. 1d). The dimeric interface is established through i) a network of hydrophobic interactions centered around the sidechain of Tyr 112, and ii) a swapped, intermolecular and anti-parallel β-sheet, referred to as the N-terminal Intermolecular Beta-Strand (NIBS). The NIBS, formed by residues 44-49, links the β-sheets of each protomer creating a continuous, intermolecular β-sheet. This structure exhibits structural similarity with a crystal structure obtained for the periplasmic domain of TolR, the ExbD counterpart in the Tol-Pal system[24]. However, this dimeric organization has not been observed in the only known ExbD$^{Ec, peri}$ structure that was shown to be monomeric[10]. This structure was solved at pH 3 and its NIBS region is disordered. Here, our experiments were performed at pH 7, and the ordered nature of the NIBS was also confirmed by relaxation measurements, as shown in Supplementary Fig. 3.

To contextualize the dimeric structure of ExbD$^{Sm, peri}$ within the whole assembly, we used our solution structure as a template to generate a model of the full ExbB-ExbD complex with AlphaFold2 (Fig. 1e, f)[26]. In the published cryo-EM structures, the ExbB-embedded helices of ExbD are not equivalent, displaying differences regarding their height and rotation towards each other as they are shifted by half a helical turn. Also it has to be noted, that in these structures only one of the two Asp 25 of ExbD is involved in hydrogen bond formation,

leaving the other Asp 25 proposed "ready" to accept a new proton for conduction[5,7]. However, in the generated model with a dimeric organization of the periplasmic domain of ExbD, the folded NIBS residues impose a structural constraint on the helices. Given that merely four loop residues lie between the two secondary structure elements (β1 starts at Asp 44 in our NMR structure and α1 ends at Leu 40 in the AlphaFold model) that could serve as hinges, the symmetry of ExbD$^{Sm, peri}$ aligns the helices in similar positions and orientations, directing both Asp 25 sidechains towards Thr 218 with a distance compatible with the formation of a hydrogen bond. Based on these observations, we postulate that this model, featuring both Asp 25 sidechains engaged in hydrogen bonds, could represent an inactive, proton-impermeable state of the channel.

## ExbD samples different conformational states

The absence of the periplasmic domain of ExbD in the cryo-EM reconstructions indicates that the dimer undergoes extensive conformational dynamics and does not simply occupy a single state as the NMR structure might imply. To characterize these conformational dynamics and to understand how the protein can fulfill its function, we explored the dynamical landscape of the periplasmic domain of ExbD. For this purpose, we analyzed ExbD$^{Sm, peri}$ by $^{15}$N chemical exchange saturation transfer (CEST) experiments, which allow for the characterization and quantification of different exchanging conformational states of a protein – with populations as sparse as less than 1 % – that are in slow exchange (lifetimes ranging from ~5 to 50 ms)[27]. The per-residue CEST profiles for the NIBS residues are characteristic of an exchange between three conformational states with a major state A (large dip) and two minor conformational states B and C (smaller dips) (Fig. 2a and Supplementary Fig. 4). We utilized the software package ChemEx[28] to fit the profiles and extract exchange parameters, including the chemical shifts of the minor conformations (Supplementary Table 4). Intriguingly, upon re-examination of 2D TROSY spectra, we were able to identify the resonances for state C as low signal-to-noise peaks (Supplementary Fig. 5). Additionally, deuterium-hydrogen exchange spectroscopy of ExbD$^{Sm, peri}$ revealed that the NIBS is more accessible to the solvent than the adjacent β-strands (Fig. 2b and Supplementary Fig. 6). This observation, in conjunction with the chemical shifts of state C, indicates that state C is an "open" state characterized by the disordered nature of the NIBS residues, as opposed to the "closed" state A, where those residues form an intramolecular β-sheet, as shown in the NMR structure. It is important to note that the dimeric structure of ExbD$^{Sm, peri}$ is maintained during this opening process as the dimer interface residues around Tyr 112 – such as Ala 106 and Phe 104 – do not feature CEST profiles indicative of exchange (Supplementary Fig. 4). We hypothesize that state B marks an intermediate state between those extrema, as its chemical shifts lie in all cases close to those of state A or state C. We note that a similar conformational exchange process is present in ExbD$^{Ec, peri}$; however, for this species, the exchange resides in the intermediate NMR timescale (μs-ms), as observed in the fingerprint spectra (Supplementary Fig. 2).

To confirm the exchange-driven order-to-disorder transition of the NIBS, we designed Val 47 mutants for both ExbD$^{Sm, peri}$ and ExbD$^{Ec, peri}$. In the structure of ExbD$^{Sm, peri}$ wild type (wt), this residue at the dimer interface occupies a hydrophobic pocket and is in close spatial proximity to the corresponding Val 47 from the other protomer within the homodimer (Fig. 2c). The mutant V47C forms a disulfide bridge locking the subunits together and, hence, mimics a pure closed state (Supplementary Fig. 7). Conversely, V47R and V47G both would lead to a dimer with an unfolded NIBS as the arginine sidechain cannot occupy the hydrophobic pocket and the glycine perturbs the stability of the NIBS – resulting in a mainly open state (Fig. 2d). These behaviors were confirmed by NMR spectroscopy, which can be briefly summarized as follows: a) The V47C mutation of ExbD$^{Sm, peri}$ shows only small chemical shift perturbations in comparison to the wt, indicating that

the main conformation of ExbD$^{Sm, peri}$ is indeed the closed state. b) The V47C mutation of ExbD$^{Ec, peri}$ results in NMR spectra absent of conformational exchange. c) The V47R and V47G mutants of both ExbD$^{Sm, peri}$ and ExbD$^{Ec, peri}$ exhibit NIBS residues with chemical shifts indicative of a disordered state (Supplementary Figs. 7–9).

Having established ExbD mutants that mainly occupy either the open (V47R/V47G) or closed state (V47C), we proceeded to study the impact of these mutations on the activity of the Has system in vivo. For that purpose, we monitored the growth of the Ton system-deficient E. coli strain K12 C600ΔhemAΔtonBΔexbBD, which requires external heme as an iron source for growth. This deficiency was compensated by two plasmids, one encoding for the entire Has operon (hasISRADEB) from S. marcescens and a second one for ExbB-ExbD$^{Sm}$. This setup allowed for a direct correlation of bacterial growth and the efficiency of heme import and, hence, reflected the correct functioning of the Has system. We also verified the assembly of the wt and mutant ExbB-ExbD complexes and confirmed that the proteins are expressed at comparable levels in the membrane, ensuring that the observed effects on bacterial growth are attributable to the specific mutations and their impact on the ExbD conformational exchange (Supplementary Fig. 10). Consequently, we could analyze the influence of ExbD adopting exclusively the open or closed state on the functioning of the Has system. Compared to the wt and the control mutant D111C (a residue far from the NIBS), V47C and V47R completely abolish growth, while V47G significantly reduces growth (Fig. 2e). These results demonstrate that neither the open state nor the closed state of ExbD alone, but rather the exchange between them, is crucial for the mechanism of the Has system, potentially reflecting distinct steps in the mechanism. Systematic mutations and in vivo photo crosslinking experiments have previously suggested the functional role of the ExbD NIBS also in E.coli[29], further emphasizing the importance of the opening and closing of ExbD as a general mechanism.

In the context of the full ExbB-ExbD complex, we hypothesize that the proton channel would predominantly occupy a main state with ExbD in a closed conformation, while a sparsely populated state would feature ExbD in an open conformation (Fig. 2f). The closed state of ExbD would thus represent the inactive, impermeable state of the ExbB-ExbD proton channel, as it should be predominantly closed to prevent continuous proton leakage. Conversely, the open state of ExbD is thought to represent the active, proton-permeable state. This arises from the transition of the NIBS residues from a folded to a disordered state. This unfolding creates an elongated loop, spanning over 20 amino acids, between the N-terminal α-helices and the periplasmic domain of ExbD, acting as a dynamic hinge. Consequently, we hypothesize that the inherent symmetry of ExbD$^{Sm, peri}$ would no longer be carried over to the organization of the N-terminal helices ensuring that both Asp 25 are no more necessarily engaged in hydrogen bonding.

## The open state of ExbD is conformationally selected upon TonB binding

To explore the interaction between HasB/TonB and ExbD, we analyzed 2D [$^1$H-$^{15}$N]-TROSY spectra of $^{15}$N-labeled HasB$^{Sm, peri}$ (residues 37-263) and $^{15}$N-labeled TonB$^{Ec, peri}$ (residues 34-239) upon addition of their unlabeled partner ExbD$^{Sm, peri}$ and ExbD$^{Ec, peri}$ respectively (Fig. 3a and Supplementary Fig. 11a). A close examination of the central region of the spectra shows the disappearance of peaks corresponding to an N-terminal IDR of HasB and TonB (Fig. 3b, c and Supplementary Fig. 11b, c), confirming their interaction with ExbD. To examine the changes on the ExbD side, we further analyzed 2D [$^1$H-$^{15}$N]-HMQC spectra of $^{15}$N-labeled ExbD$^{Sm, peri}$ and $^{15}$N-labeled ExbD$^{Ec, peri}$ with and without the respective HasB/TonB peptides that correspond to the binding regions. Peptide addition not only causes line broadening – indicative of interaction and conformational exchange between the

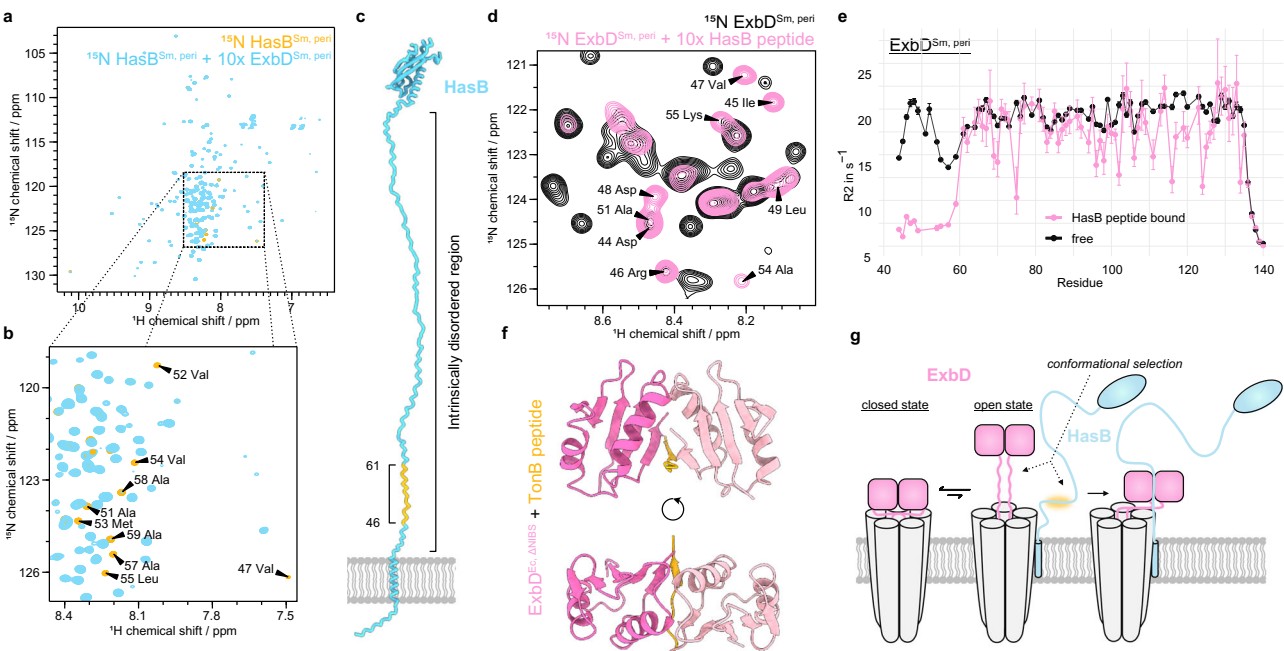

**Fig. 3 | The open state of the periplasmic domain of ExbD is conformationally selected upon binding to TonB/HasB. a, b** 2D [$^1$H-$^{15}$N]-TROSY spectra of $^{15}$N-labeled HasB$^{Sm, peri}$ alone (orange) and in the presence of unlabeled ExbD$^{Sm, peri}$ (blue). The presence of ExbD leads to the disappearance (broadening) of the peaks corresponding to the residues 46-61 of HasB$^{Sm, peri}$ indicating binding of this region to ExbD. **c** Model of HasB inserted in the inner membrane: The HasB-ExbD region of interaction (orange) is located in an intrinsically disordered region (IDR) of HasB (blue) close to the inner membrane inserted N-terminal α-helix. **d** 2D [$^1$H-$^{15}$N]-HMQC spectra of $^{15}$N-labeled ExbD$^{Sm, peri}$ without (black) and with (pink) the HasB peptide (corresponding to the binding region on the HasB side). Binding of the peptide leads to signal intensity decrease and appearance of new peaks that can be assigned to the NIBS residues. **e** R$_2$ relaxation rate measurements of the free (black) and HasB peptide bound state (pink) of $^{15}$N ExbD$^{Sm, peri}$ reveal that the binding stabilizes the

open state of ExbD$^{Sm, peri}$ characterized by the disordered nature of the NIBS residues (lower relaxation rates). This suggests a conformational selection of the open state of ExbD. Data points represent mean R$_2$ values obtained from fitting NMR data, with error bars reflecting the SD from Monte Carlo simulations performed for each residue, as detailed in the Methods. **f** Deletion of the NIBS residues of ExbDEc, peri allows for crystallization with a peptide corresponding to the binding region on the TonB side. Herein, the TonB peptide (orange) replaces the NIBS residues at the interface in between the two subunits of the ExbD dimer (pink, light pink). **g** Schematic mechanism: The open state of ExbD (pink, right side) is conformationally selected by TonB/HasB (blue) leading to a disorder-to-order transition of the IDR of TonB/HasB. The binding motif on the TonB/HasB side is highlighted in orange.

peptide bound and free state – but also leads to the emergence of novel peaks in the center of the spectrum (Fig. 3d and Supplementary Fig. 12). Interestingly, these new peaks can be assigned to the NIBS region, as they show the same chemical shifts as the NIBS residues in the minor, open conformation (State C) observed in the $^{15}$N-CEST experiment (Fig. 2a). Additionally, low R$_2$ relaxation rates confirm the disordered nature of the NIBS residues in the HasB-bound state (Fig. 3e).

Herein, we initially hypothesized that TonB/HasB might bind between the two protomers of ExbD, effectively replacing the NIBS residues at the dimer interface. This idea was driven by our observation that the NIBS residues unfold upon addition of TonB/HasB, implying a potential competition between the NIBS residues and HasB/TonB for the same binding site. To investigate this possibility, we solved the crystal structure of an ExbD construct lacking the N-terminal residues 43-60 (ExbD$^{Ec, \Delta NIBS}$) in complex with the corresponding TonB peptide at 1.5 Å resolution (Fig. 3f and Supplementary Table 5). The crystal structure confirms that the IDR of TonB undergoes a disorder-to-order transition upon binding to ExbD, replacing the NIBS residues in between the two ExbD protomers. In this arrangement, the TonB peptide forms an intermolecular β-sheet: A parallel β-sheet with the β6-strand of one protomer, and an antiparallel β-sheet with the β6-strand of the second one (Supplementary Figs. 13, 14). Corroborating these structural observations, AUC experiments of ExbD$^{Ec, peri}$ and the TonB peptide confirm that ExbD retains its dimeric structure in solution upon binding to the IDR (Supplementary Fig. 15).

Our results support that in the context of the full ExbB-ExbD complex, ExbD exchanges between a closed state and a sparsely populated open state. Upon encountering HasB/TonB, this open state is conformationally selected through the binding of HasB/TonB into the groove between the two protomers, which was previously occupied by the now-unfolded NIBS (Fig. 3g). This would mark a critical first step in the activation of the system. To further illustrate the binding mode of the ExbB-ExbD-HasB complex, we conducted modeling by AlphaFold2 using our crystal structure as a template (Supplementary Fig. 16). The model shows that the dimensions for the ExbD-HasB interaction are feasible and do not present any spatial constraints as the unfolded NIBS residues act as hinges.

## ExbD interacts transiently with the periplasmic peptidoglycan layer

Since HasB/TonB and ExbD reside in the periplasm in proximity to peptidoglycan, we explored the role of peptidoglycan in the system. For that purpose, we studied 2D [$^1$H-$^{15}$N] spectra of $^{15}$N-labeled ExbD$^{Sm, peri}$ and ExbD$^{Ec, peri}$ in the presence and absence of different peptidoglycan sacculi. Upon addition of *E. coli* peptidoglycan (Fig. 4a and Supplementary Fig. 17) some peaks loose intensity, indicating the emergence of peptidoglycan-binding induced conformational exchange. Notably, this binding is transient since fully peptidoglycan-bound ExbD would not be visible by solution NMR spectroscopy due to the megadalton size of sacculi. Mapping the decrease of peak intensities of ExbD$^{Sm, peri}$ onto the ExbD structure, shows that the dimeric interface is primarily affected by this exchange (Fig. 4b).

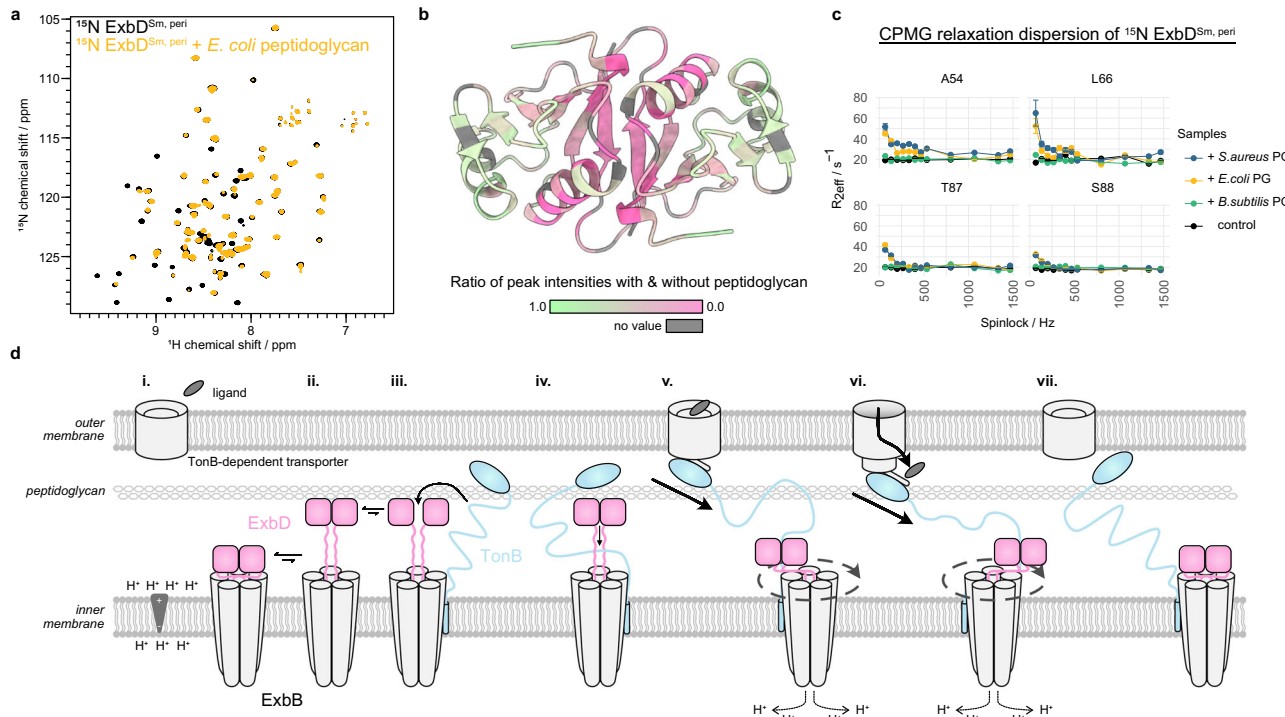

**Fig. 4 | The periplasmic domain of ExbD interacts with the peptidoglycan layer.** **a** 2D [$^1$H-$^{15}$N]-TROSY spectra of $^{15}$N ExbD$^{Sm,\ peri}$ without (black) and with (orange) *E. coli* peptidoglycan. The interaction of ExbD$^{Sm,\ peri}$ with peptidoglycan leads to the disappearance (broadening) of peaks. **b** Mapping this peak intensity loss in terms of a peak height ratio (pink to green, grey color means no value) onto the dimeric ExbD$^{Sm,\ peri}$ structure shows that the interface within the dimer experiences the largest increase in exchange contribution to the R$_2$ relaxation rate upon interaction with peptidoglycan. This suggests that possibly the dimer dissociates upon interaction. **c** This increase in R$_2$ relaxation rate due to exchange can be measured by Car-Purcell-Meiboom-Gill (CPMG) relaxation dispersion experiments and reveals that ExbD$^{Sm,\ peri}$ selectively interacts with peptidoglycan from *E. coli* (orange), *S. aureus* (blue) – they all show decaying profiles – but not *B. subtilis* (green), which shows the same flat curve as the control experiment without peptidoglycan (black). Peak intensities are plotted as mean values with error bars showing the SD, which quantifies the relative measurement error arising from noise in the NMR spectra. **d** Schematic representation of the full mechanism. In the resting state in the absence of ligand (i.), closed ExbD (pink) is in exchange with a minor open conformation (ii.). This extended, open conformation can reach the peptidoglycan layer, in interaction with which its dimer interface is loosened (iii.). This allows TonB/HasB (blue) to enter in between the ExbD monomers (iv.) and to subsequently select the open state of ExbD by binding to it (v.). Also, the C-terminal domain of TonB/HasB interacts with a ligand (grey) bound TonB-dependent transporter (TBDT). Proton motive force induced rotation of ExbD within ExbB leads to a pulling through the IDR of TonB/HasB, opening the TBDT and enabling active transport of ligands (vi.). Eventually, the pulling force becomes too large and leads to the dissociation of TonB/HasB from the TBDT, and the return to the resting state of the system (vii.). Source data are provided as a Source Data file.

Consequently, we propose that upon binding to peptidoglycan the dimeric interface of ExbD is reorganized. This reorganization would require the dissociation of the dimer. Consistent with this, the disulfide-bridged ExbD$^{Sm,\ peri}$ V47C mutant, which cannot undergo dimer dissociation, failed to interact with peptidoglycan (Supplementary Fig. 18). To identify the interaction interface on the peptidoglycan side, we further studied the interaction of ExbD$^{Sm,\ peri}$ with *Staphylococcus aureus* and *Bacillus subtilis* peptidoglycan. Compared to the peptidoglycan of *E. coli*, *S. aureus* possesses chemically different peptide stems whereas *B. subtilis* features glycans with a non-acetylated glucosamine moiety[30]. Notably, among the peptidoglycans tested here, only the one from *B. subtilis* does not interact with ExbD$^{Sm,\ peri}$, which pinpoints the interaction site on the peptidoglycan to the glycan chains (Supplementary Fig. 19). Additionally, peptidoglycan digested with mutanolysin, a muramidase, does not show any interaction with ExbD, further supporting this hypothesis (Supplementary Fig. 20). Furthermore, NMR relaxation dispersion experiments showed an increase in effective R$_2$ relaxation rates upon the addition of peptidoglycans from *E. coli* and *S. aureus*, but not from *B. subtilis* (Fig. 4c and Supplementary Fig. 21, 22).

## Discussion

We have elucidated the dimeric structure of the periplasmic domain of ExbD, which was previously unresolved in cryo-EM studies. The observed dimeric form is consistent with the two N-terminal, ExbB-integrated α1 helices of ExbD present in the cryo-EM maps. We have also demonstrated by NMR spectroscopy that the periplasmic domain of ExbD undergoes extensive conformational exchange on the milli-second timescale, sampling a primary closed state and at least one secondary open state with an unfolded NIBS region. This dynamic nature of the protein might explain its invisibility in the cryo-EM densities. The open state of ExbD is selectively bound by an IDR of the partner proteins TonB/HasB, which undergo a disorder-to-order transition as they bind between the two ExbD protomers, as evidenced by X-ray crystallography. Moreover, we have revealed that the periplasmic domain of ExbD selectively interacts with specific peptidoglycan species, most likely promoting dimer dissociation.

Taking our results together, we propose the following mechanism of action for the Ton system: A resting state is characterized by a unliganded TBDT (Fig. 4d, stage i). Within the ExbB-ExbD complex, dimeric ExbD predominantly exists in a closed conformation and a sparsely populated open conformation (Fig. 4d, stage ii.) as we have shown by NMR spectroscopy. The closed conformation likely represents the inactive, proton impermeable state of the proton channel. The sparsely populated open state of ExbD is triggered by the unfolding of the NIBS region, which subsequently reveals TonB/HasB-binding region between the ExbD protomers. Yet, even in this state, the TonB/HasB-binding region in between the ExbD dimer remains

inaccessible. This is due to structural organization of the ExbD-ExbB complex: The N-terminal α1-helices of each protomer are inserted into the ExbB channel, and the C-terminal periplasmic domain of ExbD forms a tight dimer. However, the adaptation of the sparsely populated open state following the unfolding of the NIBS region brings ExbD into close proximity with the peptidoglycan layer. Here, ExbD can bind to the glycan chains of the peptidoglycan. This binding favors the dissociation of the dimer. In this dissociated state, TonB/HasB can access in between the now-separated ExbD dimers by diffusion (Fig. 4d, stage iii.). Upon return to the dimeric structure of ExbD, the IDR of TonB/HasB is locked within the quaternary structure of the complex (Fig. 4d, stage iv.). Subsequently, TonB/HasB can bind in between the ExbD protomers at the inner membrane-facing side.

In the Tol-Pal system, dimeric TolR, a ExbD structurally homologous protein, was shown to form a complex with peptidoglycan. However, this complex formation is only possible when the β6-strand at the dimeric interface is removed yielding a different dimeric interface and organization, proposedly exposing a peptidoglycan binding motif[24,31]. This is consistent with our observation that the peptidoglycan interaction strongly affects the ExbD$^{Sm, peri}$ dimer interface. However, in our case, the presence of β6 renders the interaction with peptidoglycan more transient as we see an exchange between bound and unbound ExbD (Fig. 4a).

Upon ligand binding to the TBDT, its N-terminal extension containing the TonB box extends into the periplasm, recruiting the TonB protein. Intriguingly, molecular dynamics simulations of the IDR of TonB have shown that its Glu-Pro and Lys-Pro repeats can form a hairpin structure[17]. This configuration may serve to sterically conceal the ExbD-binding motif of TonB. We hypothesize that with its anchoring points on ExbB within the inner membrane and on the TBDT in the outer membrane, the IDR of TonB undergoes tension as both anchoring points diffuse in their respective membranes. This tension is likely to destabilize the hairpin structure, thereby unmasking the binding motif. This newly accessible motif engages with the open state of ExbD, conformationally selecting its open state. This interaction effectively stabilizes the ExbB-ExbD complex in an active, proton-permeable state (Fig. 4d, stage v.). However, in a single-target system such as the Has, the high affinity of HasB for the TBDT (nM $K_d$ versus μM for TonB-TBDTs) suggests that HasB remains associated to the TBDT even in the absence of the ligand. Here, signaling of ligand-binding is proposedly transmitted via a stabilization of the HasB-TBDT interaction by involving more polar contacts, as shown in our previous study[19]. However, it has to be noted that future research on the role and properties of the IDR of TonB/HasB will be essential to fully understand these hypothesized processes.

After the stabilization of the ExbB-ExbD complex in the active state, the PMF is translated into a rotation of ExbD within ExbB through proton translocation via the de- and reprotonation of Asp 25 of ExbD (Fig. 4d stage v.). This results in a wrapping of the intrinsically disordered linker of HasB/TonB around ExbD, exerting a pulling force on the TonB box of the TBDT. This force was shown to be sufficient in unfolding and removing the plug domain from the TBDT[32], culminating in ligand import into the periplasm (Fig. 4d, stage vi.). Finally, due to the exerted force, TonB/HasB dissociates from the TBDT. This marks the return to the resting state of the Ton system (Fig. 4d, stage vii.). This mechanism extends the wrap-and-pull model recently proposed by Ratliff et al[18].

Our study uncovers a conserved interaction between HasB/TonB and the open state of ExbD in both single- and multi-target systems. However, we showed that the transition between the open and closed states of ExbD operates at distinct rates in these systems: on the slow NMR time scale (ms) for the single-target Has system and in the intermediate time scale (μs-ms) for the multi-target Ton system. This faster rate of conformational exchange in the E. coli Ton system may

indicate a capacity for more rapid adaptation, an attribute beneficial for catering multiple TBDTs (up to 9 have been reported[33]), unlike the single-target HasB.

In homologs of ExbD of other PMF-dependent motors involved in membrane integrity (Tol-Pal system) and gliding motility (AglQRS system)[11], the NIBS residues are highly conserved, suggesting a common mode of operation involving multiple states due to the unfolding of the NIBS. This extends to the ExbD-binding motif of TonB homologs, indicating a conserved mechanism of energy transduction from the proton channel to the outer membrane (Supplementary Figs. 23, 24). We speculate that stabilizing the open state of ExbD or its homologs with a drug binding in between the protomers could not only inhibit the vital functioning of these motors but also lock the proton channel complexes into an active, proton-permeable state, creating a fatal proton leakage in the inner membrane.

## Methods

### Strains and plasmid construction
Strains, plasmids, and oligonucleotides are shown in Supplementary Table 6. To produce ExbD mutants, the pBADexbBD$_{Sm}$ or the pBAD24exbBDhis6$_{Sm}$ plasmids were respectively amplified by PCR with oligonucleotides couples ExbDV475'/ ExbDV47C, ExbDV475'/ ExbDV47G, ExbDV475'/ ExbDV47R and ExbDE1115'/ ExbDD111C. The PCR product was digested with DpnI, self-ligated and transformed in XL1-Blue. Recombinant clones were isolated and verified by sequencing.

### Protein expression and purification
Protein samples used in this study are summarized in Supplementary Table 1. All ExbD variants and TonB$^{Ec, peri}$ contain an N-terminal His-tag followed by a TEV cleavage site. The corresponding genes were synthesized and cloned by ProteoGenix into pet-30a(+) vectors, which were used to transform E.coli BL21 (DE3). For protein expression, glycerol stocks of transformed bacteria were used to inoculate 10 mL Luria-Bertani (LB) medium, which was grown for 8 h at 37 °C. In a second step, 100 mL M9 medium containing 4 g/L $^{13}$C-glucose and 1 g/ L $^{15}$NH$_4$Cl as the only carbon and nitrogen sources were added and the culture was incubated overnight at 30 °C. The next morning, 1 L fresh M9 medium was inoculated with the overnight culture to a starting optical density at 600 nm (OD) of 0.1. The bacterial cultures were grown to an OD of 0.7 at 37 °C and the expression was induced by the addition of 0.5 mM IPTG (isopropyl β-D-1-thiogalactopyranoside). Protein expression was conducted for 5 h at 37 °C (ExbD variants and TonB$^{Ec, peri}$) or for 20 h at 30 °C (HasB$_{37-263, Sm}$). For non-NMR studies, all protein expression steps were conducted in LB medium instead of M9 medium. The bacterial cells were harvested by centrifugation at 7000xg for 15 min at 37 °C, resuspended in Buffer A (20 mM sodium phosphate, 30 mM imidazole, 500 mM NaCl, pH 7.4) containing one EDTA-free protease inhibitor pill (Roche) per 25 mL and stored at −80 °C.

For expression of deuterated protein used for the detection of hydrogen bonds by NMR spectroscopy, bacteria were adapted to and grown in fully deuterated M9 medium containing $^{13}$C,D$_7$-glucose and $^{15}$ND$_4$Cl as the only carbon and nitrogen sources.

For protein purification, the bacterial cell pellets were thawed and 1.5 μL benzonase (Millipore) were added per 25 mL of cell suspension to digest nucleic acids. Cells were disrupted by sonication with a Vibracell 72405 sonicator (2 s on, 1 s off, 80% amplitude and a total time of 20 min). Insoluble cell debris was removed from the soluble protein-containing supernatant by centrifugation at 30.000xg at 4 °C for 30 min and by sterile filtration through a 0.22 μm membrane. By means of an Akta system, the lysate was loaded onto a pre-equilibrated (Buffer A: 20 mM sodium phosphate, 30 mM imidazole, 500 mM NaCl, pH 7.4) nickel affinity chromatography column (two stacked 5 mL HisTrap Chelating HP, Cytiva), washed with 5 column volumes (CV)

Buffer A and eluted with a linear gradient of 0-100 % Buffer B (20 mM sodium phosphate, 500 mM imidazole, 500 mM NaCl, pH 7.4) over 5 CV. The protein containing fractions as judged by SDS-PAGE were united and diluted (at least 10x) in TEV buffer (50 mM Tris-HCl, 1 mM dithiothreitol, pH 8.0). 1 mg of TEV protease was added per 100 mg of target protein and cleavage of the His-tag was conducted at 34 °C for 2 h. Then the mixture was loaded onto a HisTrap HP column to remove TEV-His$_6$ and un-cleaved proteins. The target protein containing fractions from the washing step were pooled, concentrated to about 1.5 mL and injected with a 2 mL loop onto a pre-equilibrated (Buffer C: 50 mM sodium phosphate, 50 mM NaCl, pH 7.0) size-exclusion chromatography column (HiLoad 16/600 Superdex 75 pg, Cytiva or HiPrep 16/60 Sephacryl S-100 HR, Cytiva) and eluted with 1.5 CV Buffer C. The fractions containing the target protein were pooled, aliquoted, flash-frozen in liquid nitrogen and stored at −20 °C. For X-ray crystallography sample preparation, Buffer D (50 mM Tris-HCl, 50 mM NaCl, pH 7.0) was used for size-exclusion chromatography.

For the His-tag lacking protein HasB$^{Sm, peri}$, the nickel affinity column steps were replaced by a cation-exchange chromatography step. For that purpose, bacterial cells were resuspended in Buffer E (50 mM Tris HCl, 100 mM NaCl, pH 8.8) after harvest. For purification, the disrupted lysate was loaded onto a pre-equilibrated (Buffer E: 50 mM Tris HCl, 100 mM NaCl, pH 8.8) cation-exchange chromatography column (SP Sepharose HP 16/60, Cytiva), washed with 10 CV Buffer E and eluted with a linear gradient of 0-100 % Buffer F (50 mM Tris HCl, 1000 mM NaCl, pH 8.8) over 15 CV. The protein containing fractions as judged by SDS-PAGE were united, concentrated and submitted to size-exclusion chromatography as mentioned above.

To acquire intermolecular NMR restraints, a sample containing 50% $^{15}$N,$^{13}$C- ExbD$^{Sm, peri}$ and 50% $^{14}$N,$^{12}$C-ExbD$^{Sm, peri}$ (50% labeled – 50% unlabeled) was prepared. For that purpose, the two batches of purified protein were mixed in equimolar quantities, denatured in 8 M Urea and subsequently refolded by dialyzing to Buffer C (3×1 L Buffer C). Fingerprint NMR spectra showed no difference between native and refolded proteins.

## Analytical ultracentrifugation (AUC)

For sedimentation velocity experiments, ExbD$^{Sm,peri}$ and ExbD$^{Ec,peri}$, and a combination of ExbD$^{Ec, peri}$ with TonB peptide were used. Concentrations of 100 μM for ExbD$^{Sm,peri}$ and ExbD$^{Ec,peri}$ were used, while for the combined experiments, ExbD$^{Ec, peri}$ was at a concentration of 100 μM and TonB peptide at 300 μM. Samples were loaded into 3 mm or 1.2 cm centerpieces and centrifuged overnight at 42000 rpm in a Beckman Coulter Optima AUC centrifuge operating with an AN60-Ti rotor. Fitting of the data using a continuous size distribution c(S) model was conducted via SEDFIT 15.1 with a confidence level (F-ratio) of 0.95. From the fit, sedimentation coefficients at zero concentration in the buffer (50 mM sodium phosphate, 50 mM NaCl, pH 7.0) could be calculated as well as molecular weights estimated[34]. Hydrodynamic parameters of the closed-state ExbD$^{Sm, peri}$ were calculated with Hydropro[35]. For the closed-state the NMR-structure determined in this study was used, the sedimentation coefficient of monomeric ExbD was calculated from PDB ID 2PFU (Monomeric ExbD structure)[10].

## Solution NMR spectroscopy and resonance assignment

All used NMR samples are summarized in Supplementary Table 2. All spectra were acquired using Topspin4 on either a 600 MHz Avance III HD or a 800 MHz Avance NEO spectrometer equipped with cryogenically cooled triple resonance $^1$H[$^{13}$C/$^{15}$N] probes (Bruker Biospin) at 293 K. For pulse calibration and setting up standard experiments NMRlib was used[36]. For spectral fingerprinting, either 2D $^{15}$N-$^1$H SOFAST-HMQC or BEST-TROSY spectra were recorded[37]. For backbone assignments, generally 3D BEST-HNCA/-HNcoCA/-HNCO/-HNCACB/-HNcoCACB and BEST-TROSY HNcaCO spectra were acquired with 25% non-uniform sampling (NUS)[38]. Sidechain

assignments were conducted by the acquisition of 3D HBHAcoNH and HCCH-TOCSY as well as 2D hbCBcgcdHD and 2D hbCBcgcdceHE spectra. For structural restraints, 3D $^{15}$N-edited NOESY-HSQC, $^{13}$C$^{aromatic}$-edited NOESY-HSQC and $^{13}$C$^{aliphatic}$-edited NOESY-HSQC, and, for intermolecular structural restraints, $^{13}$C-,$^{15}$N-filtered 3D NOESY-$^{13}$C-HSQC and $^{13}$C-,$^{15}$N-filtered 3D NOESY-$^{15}$N-HSQC spectra were recorded at a B$_0$ field strength of 800 MHz. A mixing time of 120 ms was used for all NOESY spectra. For the detection of hydrogen bonds, a 3D BEST-TROSY HNCO spectrum with a total N-CO INEPT transfer period of 133 ms was acquired[39]. Spectral data were referenced to 2,2-Dimethyl-2-silapentane-5-sulfonate (DSS), processed with NMRPipe[40] and in the case of NUS reconstructed using SMILE[41]. Visualization of spectra, peak picking and resonance assignments were conducted using CcpNmr version 2.5[42] and version 3[43].

The final resonance assignment of ExbD$^{Sm,peri}$ was deposited to the Biological Magnetic Resonance Data Bank (BMRB) under the accession code BMRB 34826 (Dimeric, periplasmic domain of ExbD).

$^{15}$N-R$_1$ and -R$_2$ relaxation rates were measured by pseudo-3D experiments ($^1$H, delay, $^{15}$N). For $^{15}$N-R$_1$ measurements the relaxation delays were 0, 20 70, 150, 200, 300, 400, 550, 700, 900, 1100, 1500, 1800 and 2000 ms in random order with a recycle delay of 5.5 s between scans. For $^{15}$N-R$_2$ measurements the relaxation delays were 0, 66.67, 133.33, 200, 266.67, 400, 533.33, 800, 1066.67, 1333.33 and 1466.67 ms in random order with a recycle delay of 4.5 s between scans. Peak heights and errors were extracted with the non-linear N-dimensional spectral modeling (nlinLS) program as part of the NMRPipe suite[40], and fitted to a monoexponential function. Hereby, the errors were estimated by running 21 Monte Carlo trials for each fit. For $^{15}$N[$^1$H] heteronuclear nuclear Overhauser effect (nOe) measurements, two HSQC spectra without (reference) and with a $^1$H saturation pulse (1 ms, 250 Hz), and with a pre-saturation delay of 3 s and a recycle delay of 10.5 s were recorded. Peak heights and errors were extracted from the two spectra with nlinLS and heteronuclear nOe values calculated as the ratio of peak heights with and without saturation pulse.

Carr-Purcell-Meiboom-Gill (CPMG) relaxation dispersion profiles were measured by BEST-TROSY pseudo-3D experiments ($^1$H, $^{15}$N spin-lock strength, $^{15}$N) at a B$_0$ field strength of 600 and 800 MHz. For these experiments the spinlock strengths were 0, 66.67, 133.33, 200, 266.67, 400, 800, 1066.67, 1333.33 and 1466.67 Hz in random order during a total relaxation delay T$_{relax}$ of 30 ms. Peak heights and errors were extracted from the spectral data with nlinLS and used to calculate the spinlock dependent R$_{2eff}$ rate following equation 1.

$$R_{2eff} = \frac{-\ln(I/I_0)}{T_{relax}} \quad (1)$$

The spinlock dependent R$_{2eff}$ rates were fitted numerically to a two-state exchange with ChemEx[28] by minimizing the target function $\chi^2$ (See equation 2).

$$\chi^2 = \sum_i \left( \frac{I_i^{expt} - I_i^{calc}}{\sigma_i^{expt}} \right)^2 \quad (2)$$

Chemical exchange saturation transfer (CEST) profiles were measured by two pseudo-3D experiments ($^1$H, $^{15}$N carrier position, $^{15}$N) with $^{15}$N spinlock strengths of 22.7 Hz and 55.9 Hz applied for a duration T$_{ex}$ of 400 ms at a B$_0$ field strength of 800 MHz. The CEST dimension had in both cases a spectral width of 2854.25 Hz, and for 22.7 Hz spinlock strength increments of 20.3 Hz and for 55.9 Hz spinlock strength increments of 40.6 Hz. The exact spinlock strength was determined by an incremented nutation experiment[44]. Peak heights and errors were extracted from the spectral data with nlinLS, and fitted to a three-state exchange with ChemEx by minimizing the target

function $\chi^2$ (See equation 2). The errors were calculated based on the scatter observed in the CEST profiles using the ChemEx function "scatter".

For the Deuterium-Hydrogen exchange experiments, we used a ExbD$^{Sm, peri}$ sample that was previously exchanged to D$_2$O. After 2 weeks, this sample was lyophilized overnight, then redissolved in Buffer C (50 mM sodium phosphate, 50 mM NaCl, pH 7.0). The first SO-FAST HMQC was recorded after 2 min 50 s with subsequent 99 time points of 2 min and 39 s. A final reference spectrum was recorded after 24 h. The peak intensities were extracted with nlinLS and fitted with fitXY.tcl of the NMRpipe suite[40] to the complement of an exponential decay (See equation 3), where A is a normalization factor depending on the maximal peak height, t the exchange time and k the exchange rate.

$$F(t) = A[1 - e^{-kt}] \qquad (3)$$

For examining the interaction between ExbD protein species and peptidoglycan, peptidoglycan sacculi from *E. coli*, *B. subtilis* or *S. aureus* (InvivoGen) were washed twice with Buffer C and added to a 100 µM protein-containing NMR sample. Simultaneously, a reference sample – maintaining the same protein concentration but devoid of peptidoglycan – was also prepared.

## NMR structure calculation of the periplasmic domain of ExbD
NMR structure calculation was performed by the ARIAweb server using the standard torsion angle simulating annealing protocol[45]. Firstly, the unambiguous intermolecular NMR restraints were used to assign peaks in the 3D $^{15}$N-edited NOESY-HSQC, $^{13}$C$^{aromatic}$-edited NOESY-HSQC and $^{13}$C$^{aliphatic}$-edited NOESY-HSQC spectra. The rationale behind this was that peak lists of the intermolecular NMR restraints cannot directly be used during the structure calculation protocol because their sparsity makes them difficult to calibrate. Additionally, further unambiguous peaks in the NOESY spectra were assigned. Torsion angles were predicted based on assigned chemical shift values using TALOS-N[46]. Secondary structure information was also used as "structural rules" in ARIA to initially prevent incorrect assignments of intermolecular NOE as intramolecular. The CcpNmr project file containing all assigned chemical shifts, NOESY peak lists, and hydrogen bonds was uploaded to the ARIAweb server. Here, 9 iterations of automatic peak assignment and calibration of the 3D $^{15}$N-edited NOESY-HSQC, $^{13}$C$^{aromatic}$-edited NOESY-HSQC, and $^{13}$C$^{aliphatic}$-edited NOESY-HSQC spectra were conducted in the following way: Manual assignments were used and flagged as reliable, diagonal peaks were filtered, and structural rules enabled. The error of chemical shifts based on the resonance assignment was used in addition to a $^1$H chemical shift tolerance of 0.04/0.02 ppm in the indirect/direct dimensions and a $^{13}$C/$^{15}$N chemical shift tolerance of 0.3 ppm. For spin diffusion correction, a molecule correlation time of 10 ns was adopted. During structure calculation, a C2 symmetry was applied along with NCS restraints[47], where the latter were used only until iteration 4. Dihedral angles derived from the TALOS-N predictions were supplied together with hydrogen bonds. In all 9 iterations, 50 structures were calculated and the 7 best structures (based on their total energy) were used for analysis. Torsion angle simulating annealing was performed with 30000 high-temperature steps, 30000 cool1 steps, and 30000 cool2 steps, and a log-harmonic potential was applied using automatic restraints weighting during the cool2 step of the simulated annealing[48]. The 10 best structures of the last iteration were refined in an explicit shell of water molecules[49]. The structure calculation protocol was iteratively repeated and some manual changes were applied to the NOE assignments in between runs. In the end, a consensus structure was calculated following the methodology reported[50]. Briefly, 20 independent ARIA runs were carried out using the same input data but different random number seeds, generating varying random initial conformations and velocities for the

molecular dynamics simulated annealing protocol. Subsequently, cross-peaks that remained active (i.e., those retaining at least one assignment possibility) were collected at the conclusion of 12 out of the 20 ARIA runs. For each active cross-peak, assignment possibilities from each individual ARIA run were combined to generate a new list of consensus distance restraints. Lastly, a single iteration ARIA run was conducted using the consensus distance restraints as input, resulting in the final consensus structure ensemble.

The final structural ensemble of ExbD$^{Sm,peri}$ and the restraints were deposited to the Protein Data Bank (PDB) under the accession code 8PEK (Structure of the dimeric, periplasmic domain of ExbD).

## Bacterial growth test
Strain *E. coli* K12 C600Δ*hemA*Δ*exbBD*Δ*tonB*(pAMhasISRADEB) was transformed with plasmid pBADexbBD$_{Sm}$ or its derivatives bearing a mutation in *exbD*. A few colonies were first inoculated in 3 mL of LB medium at 37 °C with the corresponding antibiotics, and 40 µM dipyridyl, 4 µg/mL arabinose. Once the culture reached an OD$_{600nm}$ of ca. 1.2-1.5, it was diluted and inoculated in 48 well Greiner plates, in the same medium to which was added 1 µM He-BSA, as a heme source. The initial OD$_{600nm}$ of the cultures was 0.001. Each well contained 300 µL of growth medium. Triplicates of each strain were made, and the plate was incubated at 37 °C with vigorous shaking (500 rpm) in a Clariostar Plus Microplate reader. OD$_{600nm}$ was recorded every 15 minutes for 60 hours.

## Isolation of the ExbB-ExbD$^{Sm}$ complex
Strain *E. coli* K12 C600Δ*hemA*Δ*exbBD*Δ*tonB*(pAMhasISRADEB) was transformed with the pBAD24exbBDhis6$_{Sm}$ plasmid or its variants pBAD24exbBDV47Chis6$_{Sm}$, pBAD24exbBDV47Ghis6$_{Sm}$, pBAD24exbBDV47Rhis6$_{Sm}$, and pBAD24exbBDD111Chis6$_{Sm}$. For the expression of each variant, 200 mL of culture in LB at 37 °C (in the presence of ampicillin (100 µg/mL), spectinomycin (75 µg/mL), delta-aminolevulinic acid (25 µg/mL) was induced with 40 µg/mL arabinose at OD$_{600nm}$ of 0.5 and the culture continued for 3 hours. Cells were harvested, and the pellet washed and resuspended in 100 mM Tris-HCl pH 8.0, 1 mM EDTA, in the presence of lysozyme (50 µg/mL); after one freeze-thaw cycle, the suspension was sonicated, MgSO$_4$ was added at a final concentration of 4 mM, DNase was added, and the suspension centrifuged at 16000 g for 45 minutes. The pellet was resuspended, and solubilized in 20 mM Tris-HCl pH 8.0, 100 mM NaCl, 20 mM Imidazole, 10% glycerol, 0,8% LMNG, at a final concentration of 160 OD$_{600nm}$ equivalent/mL. After 45 minutes, the suspension was centrifuged (16000 g, 45 minutes), and the supernatant incubated with Ni-Agarose. After 3 hrs, of incubation, the beads were washed 4 times with 20 mM Tris-HCl pH 8.0, 100 mM NaCl, 20 mM Imidazole, 10% glycerol, 0,005% LMNG, the bound proteins eluted in the same buffer in the presence of 200 mM Imidazole. The equivalent of 8.5 OD$_{600nm}$ was loaded on each gel lane.

## Crystallization and diffraction collection
For the structural analysis of the ExbD−TonB complex, 250 µL of 2 mM (20 mg/mL) ExbD $^{Ec, ΔNIBS}$ with 10 mM TonB peptide (KKAQPISVTMVT-PADLEPPQAKK) in Buffer G (50 mM Tris-HCl, 50 mM NaCl, pH 7.0) were prepared. Firstly, preliminary screening of crystallization conditions was performed using the vapor diffusion technique with a Mos-quitoTM nanoliter-dispensing system (TTP Labtech, Melbourn, United Kingdom) in accordance with established protocols[51]. Specifically, sitting drops were prepared by combining 400 nL of protein-peptide mixture with crystallization solutions (comprising 672 commercially available conditions) in a 1:1 ratio and then equilibrating the mixture against a 150 µL reservoir in 96-well plates (Greiner Bio-one, GmbH, Frichenhausen, Germany). The crystallization plates were subsequently maintained at 18 °C in an automated RockImager1000 (Formulatrix, Bedford, MA, United States) imager to monitor the growth of

crystals. The best crystals were obtained in 10% w/v glycerol and 3 M ammonium sulfate. The crystals were cryoprotected by soaking them into the crystallization solution supplemented with 20% glycerol as a cryoprotectant before freezing in liquid nitrogen. Subsequently, diffraction data were acquired at cryogenic temperatures (100 K) on the PROXIMA-1 beamline at the SOLEIL synchrotron facility (St Aubin, France) via MXCuBE and processed with XDS[52] through XDSME (https://github.com/legrandp/xdsme)[53].

### X-ray structure determination and refinement

AlphaFold2 multimer[54] was used to generate initial models of the ExbD-TonB peptide complex. On that basis, crystal structures were determined by molecular replacement with Phaser[55]. The final models were generated via an iterative process that involved manual model building using Coot[56] and refinement in reciprocal space with REFMAC[57] and Phenix[58] All data collection details as well as model refinement statistics are summarized in Supplementary Table 3.

The ExbD-TonB peptide complex structure was deposited to the Protein Data Bank (PDB) under the accession code 8P9R (Structure of the periplasmic domain of ExbD from *E. coli* in complex with TonB).

### Digestion of peptidoglycan

Five mg of *E. coli* peptidoglycan (InvivoGen) were washed twice with Buffer C and finally resuspended in 800 μL. An aliquot of 25 μL of a 5 mg/mL stock solution of mutanolysin (Sigma Aldrich) were added and the mixture was incubated for 2 days at 37 °C. Considering fully glycosidically-digsted non-crosslinked peptidoglycan (GlcNAc-Mur-NAc-L-Ala-D-Glu-DAP-D-Ala; molecular weight = 2060 g/mol), this yields a final concentration estimated at 2.94 mM peptidoglycan fragments. The enzymatic reaction was terminated by heating the mixture at 95 °C for 10 min. After this, the mixture was filtered through a filter with a 10 kDa cutoff to remove larger peptidoglycan fragments and any remaining mutanolysin.

### Figure creation

All depictions of protein structures were generated with ChimeraX[59].

### Reporting summary

Further information on research design is available in the Nature Portfolio Reporting Summary linked to this article.

## Data availability

NMR chemical shift assignments of ExbD$^{Sm, peri}$ are deposited at the Biological Magnetic Resonance Data Bank under the accession code BMRB 34826 (Dimeric, periplasmic domain of ExbD). The NMR protein structure of ExbD$^{Sm, peri}$ and the crystal structure of ExbD$^{Ec, ΔNIBS}$ with TonB peptide are deposited at the Protein Data Bank under the accession codes 8PEK (Structure of the dimeric, periplasmic domain of ExbD) and 8P9R (Structure of the periplasmic domain of ExbD from *E. coli* in complex with TonB) respectively. The authors declare that any other data supporting the findings of this study are available within the article and in its Supplementary Information, or from the authors upon request. Source data are provided with this paper.

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

## Acknowledgements

This work was supported by the French Agence Nationale de la Recherche (ANR Energir ANR-21-CE11-0039), the INCEPTION program "Investissement d'Avenir grant ANR-16-CONV-0005" and the Equipex CACSICE (ANR-11-EQPX-0008). We acknowledge NMR, Biophysical and Crystallography platforms of C2RT at the Institut Pasteur for their help and assistance. The 800-MHz NMR spectrometer and the optima AUC of the Institut Pasteur were partially funded by the Région Ile de France (SESAME 2014 NMRCHR grant no 4014526) and DIM one health, respectively. We thank Michael Nilges for his constant support and fruitful discussions. Molecular graphics and analyzes were performed with UCSF ChimeraX, developed by the Resource for Biocomputing, Visualization, and Informatics at the University of California, San Francisco, with support from National Institutes of Health R01-GM129325 and the Office of Cyber Infrastructure and Computational Biology, National Institute of Allergy and Infectious Diseases.

## Author contributions

N.I-P. and M.Z. conceived the study. M.Z. produced the protein samples and performed NMR measurements. M.L. assisted in the wet lab. M.Z. and N.I-P. conceived and analyzed NMR experiments. M.Z. and B.B. performed the NMR structure calculation. A.M. solved the X-ray structure. I.G.B. helped in peptidoglycan data analysis. P.D. conducted the in vivo experiments. M.Z. designed the figures and wrote the manuscript with contributions of all authors. N.I-P. guided in the writing process. All authors approved the manuscript.

## Competing interests

The authors declare no competing interests.
