## [Peer Review File · Nature Communications]

Ton Motor Conformational Switch and Peptidoglycan Role in Bacterial Nutrient UptakeREVIEWER COMMENTS

Reviewer #1 (Remarks to the Author):

Summary. The Ton complex is responsible for transducing energy from the inner membrane to the outer membrane for the gating of transporters essential for nutrient uptake and other functions. While the structures of the transporters have been solved and more recently multiple structures of subcomplexes of the Ton complex, no structure of the fully assembled Ton complex has yet been reported. As the authors here note, this is likely largely due to the dynamic nature of the complex in general, but especially for the periplasmic domain of ExbD and TonB. The authors here report the dimeric structure of ExbD and provide evidence detailing the interaction of ExbD with TonB, narrowing the interaction to a small region along the N-terminal disordered region immediately following the membrane anchor helix. The author's also provide evidence supporting the interaction of ExbD with peptidoglycan and conformational transition of the oligomeric state. Together, these results contribute to add clarity to existing gaps in the elusive mechanism of how the Ton complex partners with transporters at the outer membrane to facilitate ligand import.

General comments (in no particular order):

1. Overall the paper is well written and an excellent study that adds significant contributions to the Ton complex field of study that were previously not known. Both structural and functional studies have added clarity and important pieces of the puzzle in unraveling how the Ton complex functions in partnership with the transporters at the outer membrane. The paper contains only a few major and minor issues as noted below.

Major critiques (in no particular order):

1. The notion that multiple states (states A, B, C) of the ExbD structure are apparently entirely based on the CEST experiments for population states less than 1% (as stated). What were the % of states for the states A, B and C? Further, no structures representing states B and C are reported in the main text or the supp; it is assumed that the schematic representations from Figure 2 are based on actual structures from the NMR. If so, the authors should demonstrate these in the paper as well, since it is hard to imagine the elongated N-terminal regions of ExbD forming the ordered arrangements of state B and C without some kind of direct interaction with one another. If not interaction, then it is assumed the 10 or 20 minimized states would likely not have a common structure and would be pretty randomly distributed.
2. Similarly, reporting the structures of the two mutants here is also needed. What did the structures of the V47C and V47R/V47G mutants look like? It would be good to demonstrate that the depictions in the figures of the open and closed states are indeed a suitable representation from the NMR data.
3. Have the author done scaled modeling of a fully intact Ton complex based on their presented summary model? It seems that the region they identify that interacts with ExbD is very close to the

membrane surface and would have limited access to the binding interface of ExbD (which sits inside an ExbB pentamer), especially if ExbD is in the open state and interacting primarily at the peptidoglycan further into the periplasm. How do the authors imagine these restraints being overcome in their model? Performing scaled modeling with all known structures would help get some insight here.

4. What triggers the unfolding of the NIBS region such that TonB can bind? This interaction seems to be triggered by access of the ExbD site, rather than from TonB in response of a binding event at the outer membrane.

5. In step iii of the proposed summary model, the entire process seems stochastic and non-regulated, depending rather on the probability of rare states and events. Could the authors comment on where is the link between ligand binding and Ton activation, or does one even exist?

6. In the final step of the proposal mechanism, the authors indicate that the pulling force becomes too excessive which leads to dissociation of TonB from ExbD. Have the authors performed any pulling assays to compare the energy required to dissociate TonB from ExbD vs TonB from the TonB box of the plug domain from a transporter?

Minor critiques (in no particular order):

1. R values for the X-ray structure are a bit higher than expected for a 1.5 angstrom structure, any possible reasons for this? Additionally, percentile ranks from the validation report were mostly in the red, any specific reason for this? Doesn't indicate a wrong structure of course, but would be good to know if there were any interesting pathologies with the crystals or dataset worth note.

2. It is assumed given the resolution of the structure that all residues could be built without any ambiguity. Still, it would be good to show the density for the TonB segment in the complex structure in the supp to confirm given this is the focus on the study.

Reviewer #2 (Remarks to the Author):

ExbD is part of an inner membrane complex, which together with ExbB interacts with the transperiplasmic protein TonB and drives a family of transporters known as TonB-dependent transporters. In this work, the N-terminal motif of ExbD is explored using several methods including NMR spectroscopy. The authors present evidence that the periplasmic (soluble) portion of ExbD is a dimer, they determine the structure, and show that it is in conformational exchange between at least two states. Furthermore, they show that the interface of the dimer involves an interaction with an N-terminal sequence (termed the NIBS) that becomes disordered when released from the dimer. This dimer will also interact with a segment of TonB, also releasing the NIBS. This is interesting work, that provides some important clues to the behavior of this system, and perhaps a mechanism for transport. However, several of the conclusions reached by the authors reach do not appear to be very well justified or at least need some additional clarification for the reader. The general transport mechanism is interesting but relies on

several assumptions that have yet been demonstrated for this system. Several points that need additional clarification are listed below.

1). Line 147: The authors generate an AlphaFold structure based upon their NMR structure of the ExbD periplasmic domain and the published ExbB/ExbD structures. They find that the ExbD structure in this model is symmetric, in contrast to the cryo-EM structure. They call this symmetric structure (where both Asp 25 residues are involved in H-bonding) a closed structure. I think this conclusion needs some additional explanation. How do they know that this symmetric structure even exists? Is it possible that like ABC transporters, asymmetry is a normal state. Why do they presume that this structure closes the proton channel, and it is necessary that the structure even sample a closed state?

2). Line 202: The authors demonstrate that the periplasmic domain is in conformational exchange, and they identify one of these states as an “open” state that remains a dimer. What is the basis for assuming that the dimeric structure is maintained? This is called an open state I presume because the NIBS region is disordered. In this state, the NIBS allows the rest of the ExbD periplasmic domain to extend from ExbB. However, it is not clear how the conversion of this segment to a disordered state implies the structure is proton conductive? The authors need to provide some clarification for these conclusions.

3). Line 291: Since one can imagine that the crystal structure does not necessarily represent the protein in solution, is there evidence from AUC that ExbD retains its dimeric structure when bound to the IDR of TonB?

4). Do the authors have an idea what the affinity of ExbD is for peptidoglycan. Is it strong enough in the cell environment to permanently lock ExbD in an open configuration?

5). Line 303: It is not entirely clear what role the binding of ExbD to peptidoglycan would have. The soluble fragment of ExbD is already in conformational exchange, why do we need the peptidoglycan to promote this? If the role is to enhance interactions with TonB, the binding region from TonB is near the inner membrane and would seem to be some distance from the dimeric interface of ExbD. In this discussion, (Line 360) it is stated that the peptidoglycan interactions are promoting dissociation. What is the basis for arriving at this conclusion?

6) The authors present an interesting model for transport, but one that seems to be based upon assumptions that may or may not be true. Effectively, rotation is turned into a pulling mechanism by “winding up” or “coiling” TonB. However, it is not clear that rotation of ExbD around ExbB will necessarily result in a pulling force on TonB. This would require first that the transporter be anchored in the OM and be unable to rotate (it is probably somewhat restrained) and that torque is able to be transferred to the N-terminal motif of the transporter. This requires that the entire TonB as well as its connection to the

Ton box in the transporter acts as rigid rod. A few rotatable bonds would prevent such a mechanism from functioning. Keep in mind that the rotation model for ExbB/ExbD is at this point only a proposal, it has not been shown and rotation is only based upon a similarity to 5:2 inner membrane motors.

Minor point

Figure 3 legend: It looks like the subfigures are not labeled correctly. It appears that f) should be labeled e).

Reviewer #3 (Remarks to the Author):

This is a well written and interesting study that details the structural changes that occur in ExbD, a required component of the TonB/ExbB/ExbD complex that harnesses proton motive force to power TonB-dependent transporters (TBDTs). Both TonB and ExbD have N-terminal transmembrane helices that interact with the ExbB proton channel, followed by an intrinsically disordered region (IDR) and a small C-terminal domain that regulates the open/close of the proton channel as well as mediates interaction with the TBDT N-terminal TonBox peptide. The author extensively use NMR to study the dynamic TonB and ExbD proteins and for full transparency, I am not an expert in NMR. The authors decide to study both the *E. coli* (Ec) TonB/ExbD proteins and the HasB/ExbD system of *Serratia marcescens* (Sm). Analytical ultracentrifugation supports that both the Ec and Sm ExbD forms a dimer at neutral pH, and high quality NMR data for the Sm ExbD (authors words) provides a model of this dimer. Based on the dimer model obtained via NMR for the Sm ExbD, the authors use alphafold to postulate how the dimer may interact with and influence the ExbB pore. They speculate that the dimer of ExbD may be a part of the closed, proton impermeable form of the complex and test this by mutagenizing Val47. A V47C mutant should cross link the IDR region to the dimer and prevent opening of the complex – this mutation in vivo prevents cell growth which supports their hypothesis. The authors then examine the TonB/ExbD interaction and determine the crystal structure of the Ec ExbD (lacking the critical dimeric interface, residues 43-60) TonB IDR peptide, which demonstrates that the TonB peptide can intercalate and perhaps disrupt the dimeric ExbD complex (supported also by their NMR data that shows addition of TonB causes unfolding of the dimeric interface). Thus, the interaction of TonB (and presumably HasB) with ExbD, helps select the open ExbB pore. Finally, the authors use NMR to demonstrate that ExbD interacts with peptidoglycan and that this is dependent upon disruption of the dimer, as the V47C mutant cannot perform this binding. Overall – with the caveat that I am not an expert in NMR but rather a crystallographer with experience in TBDTs – I found this to be a great read and even after a second reading (which did help me keep the model straight and when Ec vs Sm was employed throughout), I can't really poke holes in this model or their data interpretation.

Reviewer #1

Overall the paper is well written and an excellent study that adds significant contributions to the Ton complex field of study that were previously not known. Both structural and functional studies have added clarity and important pieces of the puzzle in unraveling how the Ton complex functions in partnership with the transporters at the outer membrane. The paper contains only a few major and minor issues as noted below.

We thank the reviewer for the careful assessment of our manuscript and for recognizing the contributions of our study to the Ton system field. In the following we will address both major and minor issues.

The notion that multiple states (states A, B, C) of the ExbD structure are apparently entirely based on the CEST experiments for population states less than 1% (as stated). What were the % of states for the states A, B and C?

The specific percentages of states A, B, and C are provided in the Supplementary Information in Table S4. Here we would like to emphasize that the percentages given by CEST experiment reflect the existence and relative abundance of different conformations (major or minor) in solution and at physiological pH. However, their exact values are not of great relevance, as in our experimental conditions, the constraints from the envelope and other partners are not present.

Further, no structures representing states B and C are reported in the main text or the supp; it is assumed that the schematic representations from Figure 2 are based on actual structures from the NMR. If so, the authors should demonstrate these in the paper as well, since it is hard to imagine the elongated N-terminal regions of ExbD forming the ordered arrangements of state B and C without some kind of direct interaction with one another. If not interaction, then it is assumed the 10 or 20 minimized states would likely not have a common structure and would be pretty randomly distributed.

We thank the reviewer for this remark. The representations from Figure 2 are based on structural information from CEST experiments. This experiment is particularly suitable for characterizing and quantifying conformational states with lifetimes between 5-50 ms. Generally, domain motions and folding/unfolding events reside on this timescale (ref see main text). The only structural information that is extracted from this experiment, is the chemical shifts of the different species.

For State C, the chemical shifts, specifically those of carbon atoms that are known to be highly sensitive to the secondary structure, and, hence, allowed us to deduce secondary structures. As depicted in Figure S5 subpanel C, the NIBS residues are discernibly considered as disordered in state C.

For State B, it is not possible to extract the carbon information because of the intermediate exchange nature in the NMR time scale. However, we could identify ¹⁵N chemical shifts. Based on their values presented in Table S4, they are between states A and C, or proximal to A. This leads us to propose that State B is an intermediary between the two other states.

In essence, the CEST-derived structural takeaway is an unfolding event for the NIBS residues on the slow timescale, as corroborated by the chemical shifts. The wiggly lines representing the NIBS residues in Figure 2d symbolize their inherent disorder, not implying specific ordered configurations or

interactions. The figure 2 is a schematic representation to illustrate that the ordered-to-disordered conformational change of NIBS modulates the distance of the periplasmic domain of ExbD to the membrane. To clarify, we have added to the caption of Figure 2d (Line 185-191) *“This subpanel offers a schematic representation of the exchange dynamicsof the ExbD variants.”* and the following sentence: *“The pink rectangles depict the protomers of ExbD^{Sm, peri}. The intertwined pink lines symbolize the folded NIBS in the closed state, while the wiggly lines illustrate the disordered NIBS in the open state.”*

Similarly, reporting the structures of the two mutants here is also needed. What did the structures of the V47C and V47R/V47G mutants look like? It would be good to demonstrate that the depictions in the figures of the open and closed states are indeed a suitable representation from the NMR data.

We appreciate the reviewer's interest in obtaining detailed structures of the V47C and V47R/V47G mutants. Our approach in addressing this issue has been guided by an analysis of the NMR spectra, which, in our view, offers sufficient insights into the structural implications of these mutations.

We write in the paper in line 218-227: *“The mutant V47C is thought to form a disulfide bridge locking the subunits together and, hence, mimicking a pure closed state. Conversely, V47R and V47G both would lead to a dimer with an unfolded NIBS as the arginine sidechain cannot occupy the hydrophobic pocket and the glycine perturbs the stability of the NIBS – resulting in a mainly open state (Figure 2d). These behaviors were confirmed by NMR spectroscopy, which can be briefly summarized as follows: a) The V47C mutation of ExbD^{Sm, peri} shows only small chemical shift perturbations in comparison to the wt, indicating that the main conformation of ExbD^{Sm, peri} is indeed the closed state. b) The V47C mutation of ExbD^{Ec, peri} results in NMR spectra absent of conformational exchange. c) The V47R and V47G mutants of both ExbD^{Sm, peri} and ExbD^{Ec, peri} exhibit NIBS residues with chemical shifts indicative of a disordered state (Figure S7-9).”*

Hence, we argue on chemical shift differences between wt and mutants. For V47C it is very clear: The chemical shifts are nearly identical (Figure S7a+b) to the wt, of which we have obtained the structure in the previous chapter *“The periplasmic domain of ExbD forms a homodimer”*. For V47G/R, the general spectral fingerprint is also highly similar to the wt but the peaks corresponding to the NIBS have changed position (Figure S7a,b,c): They are now found in the center of the spectrum, an area characteristic for residues that are disordered. This means that the structures of these mutants are similar to the wt with the exception of the NIBS residues being disordered.

We have added an SDS-PAGE as the subpanel e) in Figure S7 showing the formation of the disulfide bridge in the V47C mutant. Also, we acknowledge the need for clearer explanations in the captions of Figures S7 to ensure accessibility for those less familiar with NMR. We have revised the caption (line 57-60 Supplementary file) in order to clarify:

Figure S7. a-d 2D [¹H-¹⁵N] TROSY spectra of ¹⁵N-labeled ExbD^{Sm, peri} wt, V47C, V47G and V47R variants. While the V47C mutation induces minor chemical shift perturbations (CSPs) relative to the wild type, the V47G and V47R mutations result in substantial CSPs, especially for the NIBS residues. Moreover, these latter mutations lead to a distinctive peak concentration in the spectrum's central region, characteristic of a disordered structural state. **e.** SDS-PAGE of the same variants under non-reducing and reducing (50 mM TCEP) conditions show the formation of a disulfide bridge for the V47C mutant. Thus, while the V47C variant mirrors the wt structure – with only a substituted amino acid and an added disulfide bridge – the V47G and V47R variants retain the main fold but display the NIBS residues in a disordered state.

Additionally we have revised in the main text in line 218-219: “...The mutant V47C *forms* a disulfide bridge locking the subunits together and, hence, mimics a pure closed state (Figure S7)...”

To further illustrate these NMR-results and provide a more tangible representation, we have generated AlphaFold models of these mutants. The results from these models are in line with our NMR observations and further reinforce our conclusions regarding the structural impact of the V47C, V47R, and V47G mutations. V47C is unchanged compared to the wt, whereas V47R and V47G feature a “structureless” N-terminus, which for AlphaFold models often corroborates intrinsic disorder.

V47C:

V47R:

V47G:

Have the author done scaled modeling of a fully intact Ton complex based on their presented summary model? It seems that the region they identify that interacts with ExbD is very close to the membrane surface and would have limited access to the binding interface of ExbD (which sits inside an ExbB pentamer), especially if ExbD is in the open state and interacting primarily at the peptidoglycan further into the periplasm. How do the authors imagine these restraints being overcome in their model? Performing scaled modeling with all known structures would help get some insight here.

We thank the reviewer for raising this point. We have undertaken scaled modeling of a fully intact Ton complex by using the crystal structure of the peptide-bound ExbD obtained in this work as a template. The model shows that the dimensions for the proposed interactions are feasible and do not present any spatial constraints as the unfolded NIBS residues act as hinges to allow for TonB binding. Although the residues of TonB identified to interact with ExbD are close to the inner membrane surface, our model demonstrates that it can readily access the binding interface of ExbD. We have added the model of the full ExbB-ExbD-HasB complex to the Supplementary Information (line 115-124).

The ExbD's exploration of space in the periplasm – close to the peptidoglycan – and close to the inner membrane is facilitated by the disordered nature of the NIBS residues in the open state. However, as we state in our schematic representation of the full mechanism (Figure 4d), ExbD does not interact with peptidoglycan and TonB simultaneously. We believe that the interaction with peptidoglycan merely catalyzes the dissociation of the dimer, which would allow for the intrinsically disordered region of TonB to get in between the ExbD protomers. Since the interaction with the peptidoglycan seems to be transient, ExbD would dissociate from peptidoglycan in order to bind TonB close to the inner membrane.

Figure S16 AlphaFold2 model of the full ExbB-ExbD-HasB complex, generated using the crystal structure (PDB ID 8P9R, this work) as a template. **a, b** Side views of the model (model in **b** is rotated by 180° with respect to **a**) and **c** a zoom show that the dimensions for the ExbD-HasB interaction are feasible and do not present any spatial constraints as the unfolded NIBS residues act as hinges. The ExbD pentamer is represented in white, the ExbD dimer in pink and light pink, and HasB in turquoise. The region of interaction on the HasB side as identified by NMR spectroscopy is colored in orange. For simplicity, the C-terminal IDR and globular domain of HasB (68-263) are represented by a dashed line and a turquoise ellipse, respectively.

Additionally, we have added to the main text in line 317-321:

“To further illustrate the binding mode of the ExbB-ExbD-HasB complex, we conducted modeling by AlphaFold2 using our crystal structure as a template (Figure S16). The model shows that the dimensions for the ExbD-HasB interaction are feasible and do not present any spatial constraints as the unfolded NIBS residues act as hinges.”

What triggers the unfolding of the NIBS region such that TonB can bind? This interaction seems to be triggered by access of the ExbD site, rather than from TonB in response of a binding event at the outer membrane.

Again, here, we must consider two dynamic events: 1.) **Unfolding of the NIBS that generates the open state of ExbD (without its dissociation)** 2.) **Interaction of ExbD with peptidoglycan that catalyzes the dissociation of the ExbD dimer, which would allow for TonB/HasB to get in between the two protomers.**

The unfolding of the NIBS is – as explored by CEST experiments – an inherent process. It happens without any binding partners around. However, the dissociation of the ExbD dimer to allow passage of TonB/HasB in between the protomers would require a) an unfolding of the NIBS in order to be able to explore the periplasm and b) the interaction with peptidoglycan.

As the reviewer states correctly, both these events happen independently of binding events at the outer membrane.

In step iii of the proposed summary model, the entire process seems stochastic and non-regulated, depending rather on the probability of rare states and events. Could the authors comments on where is the link between ligand binding and Ton activation, or does one even exist?

We theorize as written in the Discussion in line 411-419:

“Upon ligand binding to the TBDT, its N-terminal extension containing the TonB box extends into the periplasm, recruiting the TonB protein. Intriguingly, molecular dynamics simulations of the IDR of TonB have shown that its Glu-Pro and Lys-Pro repeats can form a hairpin structure.¹⁷ This configuration may serve to sterically conceal the ExbD-binding motif of TonB. We hypothesize that with its anchoring points on ExbB within the inner membrane and on the TBDT in the outer membrane, the IDR of TonB undergoes tension as both anchoring points diffuse in their respective membranes. This tension is likely to destabilize the hairpin structure, thereby unmasking the binding motif. This newly accessible motif engages with the open state of ExbD, conformationally selecting its open state.”

One potential regulatory mechanism might lie in this hypothesized process. Nonetheless, to gain a deeper understanding, future research will be essential, focusing on the properties of the IDR of TonB/HasB and potentially analogous systems.

We have added to the main text in line 425-427:

“However, it has to be noted that future research on the role and properties of the IDR of TonB/HasB will be essential to fully understand these hypothesized processes.”

In the final step of the proposal mechanism, the authors indicate that the pulling force becomes too excessive which leads to dissociation of TonB from ExbD. Have the authors performed any pulling assays to compare the energy required to dissociate TonB from ExbD vs TonB from the TonB box of the plug domain from a transporter?

We thank the reviewer for this remark. We did not perform any experiment to measure the pulling force between TonB-TBDT or TonB-ExbD. These experiments need a stable complex between different membrane proteins and are not easily doable. Concerning the dissociation of TonB/HasB from the TBDT, as stated in line 434-436, we hypothesize that:

“Finally, due to the exerted force, TonB/HasB dissociates from the TBDT, marking the return to the resting state of the Ton system (Figure 4d, stage vi). This complements the wrap-and-pull model proposed by Ratliff et al.¹⁸.”

R values for the X-ray structure are a bit higher than expected for a 1.5 angstrom structure, any possible reasons for this? Additionally, percentile ranks from the validation report were mostly in the

red, any specific reason for this? Doesn't indicate a wrong structure of course, but would be good to know if there were any interesting pathologies with the crystals or dataset worth note.

These relatively high R values are probably due to the peculiar way the molecules are packed in the crystal, more specifically to the fact that the peptide sits on the non-crystallographic 2-fold axis that relates ExbD dimer subunits (See new Figure S14, line 103-106).

It is assumed given the resolution of the structure that all residues could be built without any ambiguity. Still, it would be good to show the density for the TonB segment in the complex structure in the supp to confirm given this is the focus on the study.

In the revised version of the manuscript, we have included the Figure S14 in the SI showing a Polder difference map (Liebschner et al. 2017) calculated omitting the TonB peptide.

Figure S14 The green mesh corresponds to a Polder omit map¹ (contoured at 3.0σ) calculated omitting the TonB peptide (in pink). ExbD dimer chains A and B are shown in sticks and colored green and violet, respectively. Water molecules are shown as red spheres.

We have added to the main text in line 310:

“...A parallel β -sheet with the β_6 -strand of one protomer, and an antiparallel β -sheet with the β_6 -strand of the second one (Figure S13-14).”

Additionally, we have added the following reference to lines 210-211 of Supplementary Information that was used to make the figure:

Liebschner, D., Afonine, P. V., Moriarty, N. W., Poon, B. K., Sobolev, O. V., Terwilliger, T. C. & Adams, P. D. (2017). Acta Cryst. D73, 148-157.

Reviewer #2

Line 147: The authors generate an AlphaFold structure based upon their NMR structure of the ExbD periplasmic domain and the published ExbB/ExbD structures. They find that the ExbD structure in this model is symmetric, in contrast to the cryo-EM structure. They call this symmetric structure (where both Asp 25 residues are involved in H-bonding) a closed structure. I think this conclusion needs some additional explanation. How do they know that this symmetric structure even exists? Is it possible that like ABC transporters, asymmetry is a normal state. Why do they presume that this structure closes the proton channel, and it is necessary that the structure even sample a closed state?

We thank the reviewer for this remark and want to again underline that this model is a hypothesis, which is supported by our data that are consistent with the AlphaFold model.

Indeed, we postulate that the AlphaFold model generated based on our closed ExbD NMR structure “could represent an inactive, proton-impermeable state of the channel.” Firstly, we theorize that this structure exists because the folded NIBS residues impose a structural constraint on the N-terminal α -helices. There are only four loop residues between these two structural elements that could serve as hinges. Hence, the helices would be rotated symmetrically as also the NIBS strands are symmetric within the homodimer, i.e. the symmetry from the periplasmic homodimer is carried over to the N-terminal helices. In stark contrast, in the open state of ExbD where the NIBS residues are disordered, these residues can function as hinges, freeing the N-terminal α -helices from structural constraints. Secondly, if both Asp25 residues participate in hydrogen bonding, then neither is available to accept a proton for conduction. This requirement for a deprotonated Asp25, not involved in hydrogen bonding, was suggested by Biou et al. 2022. Thirdly, *in vivo* data further supports our hypothesis. The mutant that predominantly adopts the open state of ExbD (V47R) shows no cell growth (Figure 2e). This underscores the importance of ExbD's closed state and its influence on the entire system. Collectively, these findings lead us to propose the existence of this closed structure. However, it is imperative that future studies delve deeper to confirm this model.

We have revised in order to clarify in lines 155-158:

“However, in the generated model with a dimeric organization of the periplasmic domain of ExbD the folded NIBS residues impose a structural constraint on the helices. Given that merely four loop residues lie between the two secondary structure elements (Leu40 in α 1 and Asp44 in β 1) that could serve as hinges, the symmetry of ExbD^{Sm, peri} aligns the helices in similar positions and orientations, directing both Asp 25 sidechains towards Thr 218 with a distance compatible with the formation of a hydrogen bond. Based on these observations, we postulate that this model, featuring both Asp 25 sidechains engaged in hydrogen bonds, could represent an inactive, proton-impermeable state of the channel.”

Line 202: The authors demonstrate that the periplasmic domain is in conformational exchange, and they identify one of these states as an “open” state that remains a dimer. What is the basis for assuming that the dimeric structure is maintained? This is called an open state I presume because the NIBS region is disordered. In this state, the NIBS allows the rest of the ExbD periplasmic domain to extend from ExbB. However, it is not clear how the conversion of this segment to a disordered state implies the structure is proton conductive? The authors need to provide some clarification for these conclusions.

We appreciate the reviewer's questions to this point as we agree that it needs more explanation. In the submitted manuscript we had pointed out that the dimer interface is constituted by more than the NIBS, line 136-140 "...The dimeric interface is established through i) a network of hydrophobic interactions centered around the sidechain of Tyr 112, and ii) a swapped, intermolecular and anti-parallel β -sheet, referred to as the N-terminal Intermolecular Beta-Strand (NIBS). The NIBS, formed by residues 44-49, links the β -sheets of each protomer creating a continuous, intermolecular β -sheet..."

In the CEST experiments, we only see CEST profiles indicative of slow exchange for the NIBS residues. However, this behavior is not observed in other regions of the protein including other residues at the interface (please refer to the CEST profiles of residues Ala 106 or Phe 104 in Figure S4). Hence, we can conclude that the unfolding is a purely local event only involving the NIBS residues.

In addition, the transition from a folded to a disordered state of the NIBS residues would create an elongated loop, spanning over 20 amino acids between the N-terminal α -helices and the periplasmic domain of ExbD, acting as a dynamic hinge. Consequently, the inherent symmetry of ExbD^{Sm, peri} would no longer be carried over to the organization of the N-terminal helices, as it would be the case in the closed state.

We have revised the lines 208-209 to clarify:

"It is important to note that the dimeric structure of ExbD^{Sm, peri} is maintained during this opening process as the dimer interface residues around Tyr 112 – such as Ala 106 and Phe 104 – do not feature CEST profiles indicative of exchange (Supplementary Fig. 4)."

And the lines 256-262:

"In the context of the full ExbB-ExbD complex, we hypothesize that the proton channel would predominantly occupy a main state with ExbD in a closed conformation, while a sparsely populated state would feature ExbD in an open conformation (Figure 2f). The closed state of ExbD would thus represent the inactive, impermeable state of the ExbB-ExbD proton channel, as it should be predominantly closed to prevent continuous proton leakage. Conversely, the open state of ExbD is thought to represent the active, proton-permeable state. This arises from the transition of the NIBS residues from a folded to a disordered state. This unfolding creates an elongated loop, spanning over 20 amino acids, between the N-terminal α -helices and the periplasmic domain of ExbD, acting as a dynamic hinge. Consequently, the inherent symmetry of ExbD^{Sm, peri} would no longer be carried over to the organization of the N-terminal helices ensuring that both Asp25 are no more necessarily engaged in hydrogen bonding."

Line 291: Since one can imagine that the crystal structure does not necessarily represent the protein in solution, is there evidence from AUC that ExbD retains its dimeric structure when bound to the IDR of TonB?

In response to the reviewer's valid point, we have conducted additional analytical ultracentrifugation (AUC) experiments of ExbD with TonB peptide. The molecular weight of the ExbD-peptide complex in solution (24 kDa) corresponds indeed to a dimer (22 kDa) and the peptide (2.5 kDa).

We have added to the SI to the lines 108-113:

Figure S15 Sedimentation coefficient distribution obtained from analytical ultracentrifugation of ExbD^{Ec, peri} in the presence of the TonB peptide indicates that the ExbD-TonB complex maintains a dimeric organization of ExbD in solution. The red dashed lines represent estimated sedimentation coefficients for monomeric ExbD and dimeric ExbD as depicted by the cartoons (pink). The sedimentation coefficient-derived molecular weight of the complex is indicated above.

We have added to the main text (lines 310-312):

“...The crystal structure confirms that the IDR of TonB undergoes a disorder-to-order transition upon binding to ExbD, replacing the NIBS residues in between the two ExbD protomers. In this arrangement, the TonB peptide forms an intermolecular β -sheet: A parallel β -sheet with the β 6-strand of one protomer, and an antiparallel β -sheet with the β 6-strand of the second one (Figure S13). Corroborating these structural observations, AUC experiments of ExbD^{Ec, peri} and the TonB peptide confirm that ExbD retains its dimeric structure in solution upon binding to the IDR (Figure S15).”

We have added to the Methods (lines 518-521):

“For sedimentation velocity experiments, ExbD^{Sm, peri} and ExbD^{Ec, peri}, and a combination of ExbD^{Ec, peri} with TonB peptide were used. Concentrations of 100 μ M for ExbD^{Sm, peri} and ExbD^{Ec, peri} were used, while for the combined experiments, ExbD^{Ec, peri} was at a concentration of 100 μ M and TonB peptide at 300 μ M. Samples were loaded into 3 mm or 1.2 cm centerpieces and centrifuged overnight at 42000 rpm in a Beckman Coulter Optima AUC centrifuge operating with an AN60-Ti rotor. Fitting of the data using a continuous size distribution $c(s)$ model was conducted via SEDFIT 15.1 with a confidence level (F -ratio) of 0.95. From the fit, sedimentation coefficients at zero concentration in the buffer (50 mM sodium phosphate, 50 mM NaCl, pH 7.0) could be calculated as well as molecular weights estimated.³⁴ Hydrodynamic parameters of the closed-state ExbD^{Sm, peri} were calculated with Hydropro.³⁵ For the closed-state the NMR-structure determined in this study was used, the sedimentation coefficient of monomeric ExbD was calculated from PDB ID 2PFU.¹⁰”

Do the authors have an idea what the affinity of ExbD is for peptidoglycan. Is it strong enough in the cell environment to permanently lock ExbD in an open configuration?

We thank the reviewer for the insightful question. We would like to clarify that ExbD experiences two dynamic events:

1. **Unfolding of the NIBS:** This unfolding event leads to the generation of the open state of ExbD. This exchange process is inherent and does not require any interactions, and it is thoroughly described in the chapter “*ExbD samples different conformational states*”. This open state of ExbD is selected by binding to TonB/HasB.
2. **Dimer dissociation upon peptidoglycan binding:** As described in the chapter “*ExbD interacts transiently with the periplasmic peptidoglycan layer*”, this interaction leads to a reorganization of the dimer interface as “*Mapping the decrease of peak intensities of ExbD^{Sm, peri} onto the ExbD structure, shows that the dimeric interface is primarily affected by this exchange (Figure 4b).*” (line 331-332).

This is different to the unfolding of the NIBS where we only see slow-exchange for the NIBS residues but not for the entire interface. For the peptidoglycan-interaction, the entire ExbD dimer interface is affected based on the NMR data. We do not see any evidence for this dimer-monomer conformational exchange in the absence of peptidoglycan (flat CPMG relaxation dispersion curves of the control samples in Figure 4c). Therefore, we think that this dynamic event requires the presence of peptidoglycan, and we mention that “*This reorganization would require a prior dissociation of the dimer. Consistent with this, the disulfide-bridged ExbD^{Sm, peri} V47C mutant, which cannot undergo dimer dissociation, failed to interact with peptidoglycan (Figure S15).*” And, hence, we postulate that this dimer association might allow for the penetration of TonB between the ExbD dimer. To illustrate this, we have revised the figure 4d (lines 359-367).

Additionally, we have removed the word “prior” from line 334:

“This reorganization would require the dissociation of the dimer.”

In response to the affinity of ExbD for peptidoglycan: The solid nature of peptidoglycan sacculi makes it challenging to measure this affinity and we have, hence, not directly quantified this affinity. However, from the NMR measurements we can say that the binding is very transient as the signal would have completely vanished in the case of a strong binding (Figure 4a).

We think that the central point of ambiguity regarding these issues stems from the necessity for ExbD dimer dissociation induced by peptidoglycan interactions. To clarify this point, we have updated Figure 4d including a novel step iv., which explains the state of “TonB/HasB has entered in between the ExbD dimer, but not yet bound in between the protomers” (lines 359-367).

Figure 4. The periplasmic domain of ExbD interacts with the peptidoglycan layer. **a** 2D [^1H - ^{15}N]-TROSY spectra of ^{15}N ExbD^{Sm, peri} without (black) and with (orange) *E. coli* peptidoglycan. The interaction of ExbD^{Sm, peri} with peptidoglycan leads to the disappearance (broadening) of peaks. **b** Mapping this peak intensity loss in terms of a peak height ratio (pink to green, grey color means no value) onto the dimeric ExbD^{Sm, peri} structure shows that the interface within the dimer experiences the largest increase in exchange contribution to the R_2 relaxation rate upon interaction with peptidoglycan. This suggests that possibly the dimer dissociates upon interaction. **c** This increase in R_2 relaxation rate due to exchange can be measured by Car-Purcell-Meiboom-Gill (CPMG) relaxation dispersion experiments and reveals that ExbD^{Sm, peri} selectively interacts with peptidoglycan from *E. coli* (orange), *S. aureus* (blue) – they all show decaying profiles – but not *B. subtilis* (green), which shows the same flat curve as the control experiment without peptidoglycan (black). **d** Schematic representation of the full mechanism. In the resting state in the absence of ligand (i.), closed ExbD (pink) is in exchange with a minor open conformation (ii.). This extended, open conformation can reach the peptidoglycan layer, in interaction with which its dimer interface is loosened (iii.). This allows TonB/HasB (blue) to enter in between the ExbD monomers (iv.) and to subsequently select the open state of ExbD by binding to it (v.). Also, the C-terminal domain of TonB/HasB interacts with a ligand (grey) bound TonB-dependent transporter (TBDT). Proton motive force induced rotation of ExbD within ExbB leads to a pulling through the IDR of TonB/HasB opening the TBDT and enabling active transport of ligands (vi.). Eventually, the pulling force becomes too large and leads to the dissociation of TonB/HasB from the TBDT, and the return to the resting state of the system (vii.).

Additionally, we have modified and expanded in the Discussion (line 386-403):

“...Taking our results together, we propose the following mechanism of action for the Ton system: A resting state is characterized by a unliganded TBDT (Figure 4d, stage i). Within the ExbB-ExbD complex, dimeric ExbD predominantly exists in a closed conformation and a sparsely populated open conformation (Figure 4d, stage ii.) as we have shown by NMR spectroscopy. The closed conformation likely represents the inactive, proton impermeable state of the proton channel. The sparsely populated open state of ExbD is triggered by the unfolding of the NIBS region, which subsequently reveals TonB/HasB-binding region between the ExbD protomers. Yet, even in this state, the TonB/HasB-binding region in between the ExbD dimer remains inaccessible. This is due to structural organization of the ExbD-ExbB complex: The N-terminal α 1-helices of each protomer are inserted into the ExbB channel, and the C-terminal periplasmic domain of ExbD forms a tight dimer. However, the adaptation of the sparsely populated open state following the unfolding of the NIBS region brings ExbD into close

proximity with the peptidoglycan layer. Here, ExbD can bind to the glycan chains of the peptidoglycan. This binding favors the dissociation of the dimer. In this dissociated state, TonB/HasB can access in between the now-separated ExbD dimers by diffusion (Figure 4d, stage iii.). Upon return to the dimeric structure of ExbD, the IDR of TonB/HasB is locked within the quaternary structure of the complex (Figure 4d, stage iv.). Subsequently, TonB/HasB can bind in between the ExbD protomers at the inner membrane-facing side...”

Line 303: It is not entirely clear what role the binding of ExbD to peptidoglycan would have. The soluble fragment of ExbD is already in conformational exchange, why do we need the peptidoglycan to promote this? If the role is to enhance interactions with TonB, the binding region from TonB is near the inner membrane and would seem to be some distance from the dimeric interface of ExbD. In this discussion, (Line 360) it is stated that the peptidoglycan interactions are promoting dissociation. What is the basis for arriving at this conclusion?

We apologize that this point was not clear. We have answered and revised regarding this point under the previous comment. Shortly: We propose that the role of peptidoglycan binding is to induce dimer-monomer exchange of ExbD so that TonB/HasB can get in between the ExbD dimer. This is a necessity as the HasB/TonB-binding site is found at the inner membrane-facing side of ExbD.

The basis to arrive at the conclusion is twofold:

1. **The PG-induced exchange affects mostly the dimer interface:** We write in lines 331-332 “Mapping the decrease of peak intensities of ExbD^{Sm, peri} onto the ExbD structure, shows that the dimeric interface is primarily affected by this exchange (Figure 4b). Consequently, we propose that upon binding to peptidoglycan the dimeric interface of ExbD is reorganized. This reorganization would require a prior dissociation of the dimer. Consistent with this, the disulfide-bridged ExbD^{Sm, peri} V47C mutant, which cannot undergo dimer dissociation, failed to interact with peptidoglycan (Figure S18).”
2. **Literature on TolR:** TolR (ExbD homologous protein) was shown to bind peptidoglycan. However, the peptidoglycan-binding structure of TolR differs from its non-peptidoglycan binding structure, i.e. to accommodate such a reorganization of the dimer, a prior dimer to monomer dissociation is required. We write in lines 404-410 “In the Tol-Pal system, dimeric TolR, a ExbD structurally homologous protein, was shown to form a complex with peptidoglycan. However, this complex formation is only possible when the $\beta 6$ -strand at the dimeric interface is removed yielding a different dimeric interface and organization, proposedly exposing a peptidoglycan binding motif.^{24,31} This is consistent with our observation that the peptidoglycan-interaction strongly affects the ExbD^{Sm, peri} dimer interface. However, in our case, the presence of $\beta 6$ renders the interaction with peptidoglycan more transient as we see an exchange between bound and unbound ExbD (Figure 4a).”

The authors present an interesting model for transport, but one that seems to be based upon assumptions that may or may not be true. Effectively, rotation is turned into a pulling mechanism by “winding up” or “coiling” TonB. However, it is not clear that rotation of ExbD around ExbB will

necessarily result in a pulling force on TonB. This would require first that the transporter be anchored in the OM and be unable to rotate (it is probably somewhat restrained) and that torque is able to be transferred to the N-terminal motif of the transporter. This requires that the entire TonB as well as its connection to the Ton box in the transporter acts as rigid rod. A few rotatable bonds would prevent such a mechanism from functioning. Keep in mind that the rotation model for ExbB/ExbD is at this point only a proposal, it has not been shown and rotation is only based upon a similarity to 5:2 inner membrane motors.

We thank the reviewer for this remark. Like the reviewer we do not believe that a) the transporter cannot rotate in the outer membrane and b) that no bond rotations happen in the IDR of both TonB and HasB.

In the paper, we built as stated onto the wrap-and-pull model proposed by Ratcliff et al. 2022: Upon rotation of ExbD within ExbB, the IDRs of HasB/TonB are wrapped around the complex as they are a) attached to ExbB via their N-terminal α -helix and b) locked in between the ExbD dimer. This wrapping – similar to the translation of motion of a fishing rod: rotation of the reel is translated into pulling force – is, hence, translated into a pulling. Based on our structures and data, we think that this is the most probable mechanism of action.

Figure 3 legend: It looks like the subfigures are not labeled correctly. It appears that f) should be labeled e).

We thank the reviewer for this observation and have corrected the labeling accordingly.

Reviewer #3

This is a well written and interesting study that details the structural changes that occur in ExbD, a required component of the TonB/ExbB/ExbD complex that harnesses proton motive force to power TonB-dependent transporters (TBDTs). Both TonB and ExbD have N-terminal transmembrane helices that interact with the ExbB proton channel, followed by an intrinsically disordered region (IDR) and a small C-terminal domain that regulates the open/close of the proton channel as well as mediates interaction with the TBDT N-terminal TonBox peptide. The author extensively use NMR to study the dynamic TonB and ExbD proteins and for full transparency, I am not an expert in NMR. The authors decide to study both the *E. coli* (Ec) TonB/ExbD proteins and the HasB/ExbD system of *Serratia marcescens* (Sm). Analytical ultracentrifugation supports that both the Ec and Sm ExbD forms a dimer at neutral pH, and high quality NMR data for the Sm ExbD (authors words) provides a model of this dimer. Based on the dimer model obtained via NMR for the Sm ExbD, the authors use alphafold to postulate how the dimer may interact with and influence the ExbB pore. They speculate that the dimer of ExbD may be a part of the closed, proton impermeable form of the complex and test this by mutagenizing Val47. A V47C mutant should cross link the IDR region to the dimer and prevent opening of the complex – this mutation in vivo prevents cell growth which supports their hypothesis. The authors then examine the TonB/ExbD interaction and determine the crystal structure of the Ec ExbD (lacking the critical dimeric interface, residues 43-60) TonB IDR peptide, which demonstrates that the TonB peptide can intercalate and perhaps disrupt the dimeric ExbD complex (supported also by their NMR data that shows addition of TonB causes unfolding of the dimeric interface). Thus, the interaction of TonB (and presumably HasB) with ExbD, helps select the open ExbB pore. Finally, the authors use NMR to demonstrate that ExbD interacts with peptidoglycan and that this is dependent upon disruption of the dimer, as the V47C mutant cannot perform this binding. Overall – with the caveat that I am not an expert in NMR but rather a crystallographer with experience in TBDTs – I found this to be a great read and even after a second reading (which did help me keep the model straight and when Ec vs Sm was employed throughout), I can't really poke holes in this model or their data interpretation.

We thank the reviewer for their thorough and positive feedback on our study.

REVIEWERS' COMMENTS

Reviewer #1 (Remarks to the Author):

The authors have adequately addressed previous critiques by providing additional explanations, figures, and modifications as necessary. This reviewer thanks the authors for their careful responses and changes to improve the manuscript. Congratulations on a nice overall study that provides important details about how the Ton complex functions.

Reviewer #2 (Remarks to the Author):

The revised manuscript makes an interesting and unique contribution to the literature on the molecular mechanism by which the TonB system functions. A range of experiments provide the basis for a model where conformational exchange and ExbD/TonB interactions help couple the inner membrane pmf to outer membrane transporters. The authors have largely addressed my concerns with the original submission, and the work represents an important contribution to this field.

Reviewer #3 (Remarks to the Author):

I think the authors did a great job of addressing all of the reviewer concerns. This is a really interesting story and rigorous work.

REVIEWERS' COMMENTS

Reviewer #1 (Remarks to the Author):

The authors have adequately addressed previous critiques by providing additional explanations, figures, and modifications as necessary. This reviewer thanks the authors for their careful responses and changes to improve the manuscript. Congratulations on a nice overall study that provides important details about how the Ton complex functions.

We thank the reviewer for their highly positive reply, and for the work and time invested in the thorough reviewing process.

Reviewer #2 (Remarks to the Author):

The revised manuscript makes an interesting and unique contribution to the literature on the molecular mechanism by which the TonB system functions. A range of experiments provide the basis for a model where conformational exchange and ExbD/TonB interactions help couple the inner membrane pmf to outer membrane transporters. The authors have largely addressed my concerns with the original submission, and the work represents an important contribution to this field.

We thank the reviewer for their highly positive reply, and for the work and time invested in the thorough reviewing process

Reviewer #3 (Remarks to the Author):

I think the authors did a great job of addressing all of the reviewer concerns. This is a really interesting story and rigorous work.

We thank the reviewer for their highly positive reply, and for the work and time invested in the thorough reviewing process